# Nabla-R2D3: Effective and Efficient 3D Diffusion Alignment with 2D Rewards

**Qingming Liu**[1,*]    **Zhen Liu**[1,*,†]    **Dinghuai Zhang**[2]    **Kui Jia**[1]

[1]The Chinese University of Hong Kong, Shenzhen    [2]Microsoft Research

[*]Equal contribution    [†]Corresponding author

Project page: nabla-r2d3.github.io

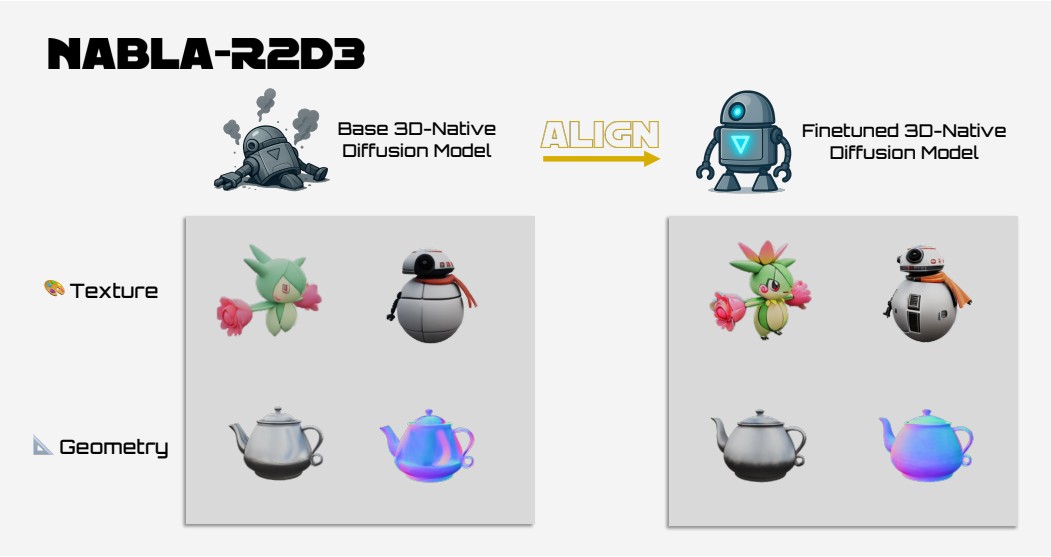

Figure 1: Our **Nabla-R2D3** can efficiently and robustly finetune 3D-native diffusion models with differentiable reward models learned from human preferences on appearance, geometry and many other attributes.

## Abstract

Generating high-quality and photorealistic 3D assets remains a longstanding challenge in 3D vision and computer graphics. Although state-of-the-art generative models, such as diffusion models, have made significant progress in 3D generation, they often fall short of human-designed content due to limited ability to follow instructions, align with human preferences, or produce realistic textures, geometries, and physical attributes. In this paper, we introduce **Nabla-R2D3**, a highly effective and sample-efficient reinforcement learning alignment framework for 3D-native diffusion models using 2D rewards. Built upon the recently proposed Nabla-GFlowNet method, which matches the score function to reward gradients in a principled manner for reward finetuning, our **Nabla-R2D3** enables effective adaptation of 3D diffusion models using only 2D reward signals. Extensive experiments show that, unlike vanilla finetuning baselines which either struggle to converge or suffer from reward hacking, **Nabla-R2D3** consistently achieves higher rewards and reduced prior forgetting within a few finetuning steps.

39th Conference on Neural Information Processing Systems (NeurIPS 2025).

# 1 Introduction

Recent advances in 3D generative models have enabled non-experts to produce batches of 3D digital assets at low cost for downstream tasks such as gaming, film, and robotics simulation. However, these assets are rarely production-ready and often require post-processing due to low visual fidelity, suboptimal geometry, ethical biases, and poor instruction following (in text-to-3D setups). These issues largely stem from training datasets that diverge from human preferences and 3D constraints.

A common solution is to first obtain a reward model that reflects human preferences and later perform reward finetuning on the generative model—a process commonly known as reinforcement learning from human feedback (RLHF). Originally developed for aligning autoregressive language models, reward finetuning has also been successfully applied to diffusion models, which are among the most popular generative models in the vision domain. It is therefore natural to consider applying similar techniques to 3D diffusion models.

However, extending RLHF to 3D diffusion models is non-trivial due to the lack of high-quality 3D reward models. In 2D settings, reward finetuning typically relies on 2D reward models; by analogy, 3D finetuning would require 3D reward models. Yet collecting diverse and high-quality 3D data remains a longstanding challenge, making it difficult to build reward models that reflect human preferences on attributes such as aesthetics, geometry, instruction following, and physical plausibility.

Inspired by the success of the "lifting from 2D" approach for 3D generation, where one optimizes a 3D shape such that each view is of high likelihood under a pretrained 2D generative model, we propose to similarly finetune 3D-native diffusion models [46, 61, 38, 23, 24, 44] with 2D reward models. Using a 3D-native model, we can sample different camera views and perform the lifting operation in an amortized fashion across training instances, rather than optimizing each object individually. However, sampling from 2D views yields high variance during optimization and can lead to instability and overfitting of 3D-native diffusion models. In light of a state-of-the-art reward finetuning method called Nabla-GFlowNet [25], which proves to be highly robust, efficient and effective on 2D diffusion models, we propose **Nabla-R2D3** (short for **R**eward from **2**D for **D**iffusion Alignment in **3**D via **Nabla**-GFlowNet), which adapts this method to finetune 3D-native diffusion models with 2D rewards. We empirically show that our method produces 3D-native models that are better aligned with human preferences and avoid major artifacts, such as unexpected floaters, commonly produced by prior methods. Furthermore, we demonstrate the effectiveness of different 2D reward models for aligning 3D-native diffusion models for different preferences.

We summarize our major contributions in this paper below:

- To the best of our knowledge, **Nabla-R2D3** is the first method to effectively and robustly align 3D-native diffusion models with human preferences using only 2D reward models.

- We demonstrate several examples of 2D reward models, including appearance-based and geometry-based ones, for aligning 3D-native generative models on different attributes.

- Our extensive experiments show that, compared to the proposed vanilla reward finetuning baselines, our **Nabla-R2D3** can effectively, efficiently and robustly finetune 3D-native generative models from 2D reward models with better preference alignment, better text-object alignment and fewer geometric artifacts.

# 2 Related Work

**Reward Finetuning of Diffusion Models.** The earliest attempt at reward finetuning for diffusion models, named DDPO [2], views the denoising process of diffusion models as trajectory sampling in a Markov decision process (MDP), in which each state is a tuple of a noisy image and the corresponding time step; a sampled trajectory starts from a random Gaussian noise at time $T$ and, through the iterative stochastic denoising process, reaches a sample image at time $0$. With this MDP defined, DDPO leverages the classical policy gradient method in reinforcement learning (RL) to finetune diffusion models. As reward models are typically learned with neural nets and thus differentiable, DDPO does not effectively leverage the available first-order information in the reward model. To address this issue, methods like ReFL [48] and DRaFT [4] treat each sample trajectory as a deep computational graph and, by assigning reward values to the states, use back-propagation to directly optimize model parameters with respect to reward signals in an end-to-end fashion. While efficient in practice, these

methods lack probabilistic grounding—their training objectives are not designed to approximate the reward-weighted distribution—and thus tend to overfit to the reward model. A parallel path is to adopt stochastic optimal control (SOC) that frames the alignment objective as an optimal control problem. The SOC approach in theory may achieve ideal results, but the proposed methods [42] so far are either ineffective or computationally expensive. Recently, new RL-based diffusion finetuning methods are proposed in the framework of generative flow networks [1, 58, 55, 27, 30, 57, 56, 59, 52] (or GFlowNets in short) that builds generative models on a directed acyclic graph to generate samples according to a reward distribution. The resulting finetuning methods, albeit derived in GFlowNet language, are also deeply rooted in (and in many cases equivalent to) soft Q-learning [9] in reinforcement learning. These new RL methods are constructed from first principles and are shown to be effective and efficient in creating finetuned diffusion models that generate diverse samples.

**3D Generation via "Lifting from 2D".** Due to the scarcity of 3D data, several works propose generating 3D shapes with only 2D signals. One early work in this direction is Dream Field [14], which initializes a 3D shape and optimizes the shape with CLIP scores on images rendered from randomly sampled camera views of the shape. Since the CLIP model is not a generative model, it is hard, if not impossible, to yield detailed appearance and geometry in generated 3D shapes. Following Dream Field, DreamFusion [31] was proposed to use score distillation sampling (SDS) that replaces CLIP scores with diffusion losses on rendered views. Such an approach is used not only for 3D object synthesis from scratch with 2D models, but also for texture generation on known geometry [36], 3D object synthesis with video generative models [15] and so on. However, the lifting approach is highly unstable and requires extensive hyperparameter tuning [31]. Another issue is the famous Janus problem that implausible 3D objects may be generated even if most of the 2D views are reasonable—demonstrating the issue of the lack of 3D priors. In addition, these lifting-based 3D generation methods take a long time to sample only one single object and are less suitable for downstream tasks due to high computational cost. Our proposed method is free from these issues because it directly works on and infers from 3D-native diffusion models.

**Alignment in 3D Generation.** Apart from the per-sample alignment with SDS-based "lifting from 2D" approaches [51, 63, 13], probably the methods most relevant to ours are MVReward [43] and Carve3D [47], both of which finetune a multi-view diffusion model with a separate multi-view-based reward model. These methods assume a multi-view representation of 3D objects, a representation that does not guarantee 3D consistency. Moreover, since this representation does not encode ground-truth normal or depth maps, external estimators must be employed to improve geometry using geometric rewards. Another line of 3D alignment is through direct preference optimization (DPO) [35, 63], where a model is finetuned on a preference dataset and without any explicit reward model. While DPO is conceptually simple, alignment with a reward model is generally better [29] and applies to scenarios where we have analytical and/or expert-designed reward models. There is a recent trend of post-training alignment with test-time scaling [16, 28] to filter undesired samples or dynamically adjust sampling strategy during inference. Such a strategy is applied to 3D generation [7], but it is typically costly and do not leverage the gradient information in reward models.

## 3 Preliminaries

### 3.1 Diffusion models and RL-based finetuning

Diffusion models are a powerful class of generative models that generate samples through sequential denoising process. To be specific, it typically starts from time $T$ with a point $x_T$ sampled from a standard Gaussian distribution, and gradually generate cleaner samples through a learned backward process $p(x_{t-1}|x_t)$ until it obtains the final sample $x_0$. The backward process $p(x_{t-1}|x_t)$ is trained to match a forward process $q(x_t|x_{t-1})$, typically set as a simple Gaussian distribution. For instance, in a popular diffusion model DDPM [11], the corresponding noising process is $q(x_{t+1}|x_t) = \mathcal{N}(\sqrt{\alpha_{t+1}/\alpha_t}x_t, \sqrt{1 - \alpha_{t+1}/\alpha_t}I)$ and $q(x_t|x_0) = \mathcal{N}(\sqrt{\alpha_t}x_t, \sqrt{1 - \alpha_t}I)$ with a noise schedule $\{\alpha_t\}_t$. To train a DDPM model on a dataset $\mathcal{D}$, we use the score matching loss:

$$\mathbb{E}_{t\sim\text{Uniform}(\{1,...,T\}),\epsilon\sim\mathcal{N}(0,I),x_T\sim\mathcal{D}} \; w(t)\|\epsilon_\theta(\sqrt{\alpha_t}x_T + \sqrt{1 - \alpha_t}\epsilon, t) - \epsilon\|^2, \tag{1}$$

where $w(t)$ is a weighting scalar, and $\epsilon_\theta(x_t, t)$ is a neural net that predicts the noise vector $\epsilon$ from $x_t$. The stochastic sequential denoising process can be treated as a Markov decision process (MDP) in which $(x_t, t)$ is a state, $(x_T, T)$ is the initial state, $(x_0, 0)$ is the final state, $p(x_t|x_{t+1})$ is the transition function. With such an MDP defined, one can align the underlying diffusion model with any reinforcement learning algorithm [2] and optimize some terminal reward $R(x_0)$.

## 3.2 Diffusion alignment via gradient-informed RL finetuning

To preserve prior in the pretrained model $p_{\text{base}}(x_t|x_{t+1})$, a typical finetuning objective is to match the "tilted" reward distribution $p_{\text{base}}(x)R^{\beta}(x)$ where $\beta$ controls the amount of prior information in the finetuned model. With the MDP defined in Sec. 3.1, it is shown that we may collect on-policy trajectories $\{(x_T, ..., x_0)_k\}_{k=1}^{K}$ from the finetuned model $p_{\theta}(x_t|x_{t+1})$ (where $K$ is the batch size) and optimize some RL objective. A recent method called Nabla-GFlowNet shows that we may efficiently and robustly finetune a diffusion model with "score-matching-like" consistency losses:

$$\mathcal{L}_{\text{forward}}(x_{t-1:t}) = \left\| \nabla_{x_{t-1}} \log \tilde{p}_{\theta}(x_{t-1}|x_t) - \gamma_t \beta \overline{\nabla} \left[ \nabla_{x_{t-1}} \log R(\hat{x}_{\theta}(x_{t-1})) \right] - g_{\phi}(x_{t-1}) \right\|^2, \quad (2)$$

$$\mathcal{L}_{\text{reverse}}(x_{t-1:t}) = \left\| \nabla_{x_t} \log \tilde{p}_{\theta}(x_{t-1}|x_t) + \gamma_t \beta \overline{\nabla} \left[ \nabla_{x_t} \log R(\hat{x}_{\theta}(x_t)) \right] + g_{\phi}(x_t) \right\|^2, \quad (3)$$

and the terminal loss $\mathcal{L}_{\text{terminal}}(x_0) = \|g_{\phi}(x_0)\|^2$, where $\log \tilde{p}_{\theta} = \log p_{\theta} - \log p_{\text{base}}$ represents the log-density ratio between the finetuned and base models., $\hat{x}_{\theta}(x_t) = (x_t - \sigma_t \epsilon_{\theta}(x_t))/\alpha_t$ is the expected one-step prediction of $x_0$ (i.e., $\mathbb{E}[x_0 \mid x_t]$) under the finetuned model, $\epsilon_{\theta}(x_t)$ is the noise prediction network of the finetuned model ($\alpha_t$ and $\sigma_t$ denote the signal and noise scales from the forward process). $\overline{\nabla}$ is the stop-gradient operation, $\beta$ controls the relative strength of the reward with respect to the prior of the pretrained model and $\gamma_t$ is the decay factor of the guessed gradient.

The total loss follows (where $w_B$ is a non-negative weighting scalar):

$$\mathcal{L}_{\text{total}} = \mathbb{E}_{x_T \sim \mathcal{N}(0,I),(x_{T-1},...,x_0) \sim p_{\theta}(x_t|x_{t+1})} \left[ \mathcal{L}_{\text{terminal}} + \sum_t (\mathcal{L}_{\text{forward}} + w_B \mathcal{L}_{\text{reverse}}) \right]. \quad (4)$$

This loss is originally derived within the framework of GFlowNet—a generative model with the MDP defined on a directed acyclic graph in which the terminal states are sampled with probability proportional to the corresponding rewards. It is shown [25] that this special set of losses is indeed equivalent to a gradient version of the soft Q-learning loss [9] in the reinforcement learning literature.

# 4 Method

## 4.1 Efficient and Robust 3D Diffusion Alignment with 2D Rewards

Suppose that we have a 2D reward model $R(x_0)$ that maps some image $x_0$ to the corresponding reward. We adopt a common assumption in the text-to-3D literature: the 3D reward model can be derived from a 2D reward model via multi-view rendering.

$$\log R_{\text{3D}}(z_0) = \mathbb{E}_{c \sim \mathcal{C}} \log R(h(z_0, c)), \quad (5)$$

where $z_0$ is a clean 3D shape, $h(z_0, c)$ is the rendering function that maps $z_0$ to the image with camera pose $c$ sampled from a set of camera poses $\mathcal{C}$. Notice that if $R(\cdot)$ is approximated with the negative diffusion loss with a pretrained 2D diffusion model, the 3D reward is basically the score distillation sampling (SDS) objective in DreamFusion [31] (with slight differences in implementation):

$$\log R_{\text{3D, SDS}}(z_0) = - \mathbb{E}_{\epsilon \sim \mathcal{N}(0,I), t \sim \text{Uniform}(\{1,2,...,T\}), c \sim \mathcal{C}, x_t = \alpha_t h(z_0,c) + \sigma_t \epsilon} w_t \|\epsilon_{\text{2D}}(x_t) - \epsilon\|^2. \quad (6)$$

where $\epsilon_{\text{2D}}$ is the $\epsilon$-prediction network of the 2D diffusion model and $w_t$ is some weight factor.

Based on the above assumption, we derive the following $\nabla$-DB losses for finetuning 3D diffusion models on 2D reward models (with the terminal loss staying the same as $L_{\text{terminal}}(z_0) = \|g_{\phi}(z_0)\|^2$):

$$L_{\text{forward}}(z_{t-1:t}) = \mathbb{E}_{c \sim \mathcal{C}} \left\| \nabla_{z_{t-1}} \log \tilde{p}_{\theta}(z_{t-1}|z_t) - \gamma_t \beta \overline{\nabla} \left[ \nabla_{z_{t-1}} \log R(h(\hat{z}_{\theta}(z_{t-1}), c)) \right] - g_{\phi}(z_{t-1}) \right\|^2, \quad (7)$$

$$L_{\text{reverse}}(z_{t-1:t}) = \mathbb{E}_{c \sim \mathcal{C}} \left\| \nabla_{z_t} \log \tilde{p}_{\theta}(z_{t-1}|z_t) + \gamma_t \beta \overline{\nabla} \left[ \nabla_{z_t} \log R(h(\hat{z}_{\theta}(z_t), c)) \right] + g_{\phi}(z_t) \right\|^2. \quad (8)$$

Empirically, we find that the following approximate loss alone works well:

$$L_{\text{approx}}(z_{t-1:t}) = \mathbb{E}_{c \sim \mathcal{C}} \left\| \nabla_{z_{t-1}} \log \tilde{p}_{\theta}(z_{t-1}|z_t) - \gamma_t \beta \overline{\nabla} \left[ \nabla_{z_{t-1}} \log R(h(\hat{z}_{\theta}(z_{t-1}), c)) \right] \right\|^2, \quad (9)$$

with which we assume that our educated guess of the "reward gradient" is accurate and therefore there is no need to learn the correction term $g_{\phi}$ with the reverse-direction loss anymore. We use this simple loss throughout the rest of the paper.

## 4.2 Practical 2D Reward Models for 3D Native Diffusion Models

**2D rewards from appearance.** It has been shown that RGB appearance in 2D views alone can support high-quality 3D generation, as demonstrated by lifting-from-2D methods [31]. We follow recent methods in SDS-based reward-guided 3D generation and consider the following reward models for finetuning 3D-native diffusion models: 1) *Aesthetic Score* [17], trained on the LAION-Aesthetic dataset [17] to measure the aesthetics of images, and 2) *HPSv2* [45], trained on HPDv2 dataset [45] to measure general human preferences over image quality and text-image alignment.

**2D rewards from geometry.** In many cases, multiview RGB information can still fall short in creating fine-grained and 3D-consistent geometric details due to the lack of sufficient 3D priors or regularization. Inspired by the recent progress in single-view depth and normal estimation, we propose to explicitly encourage the consistency between rendered RGB images and the 3D geometric predictions inferred from those images. In our experiments, we employ state-of-the-art normal map estimators and take as the reward the inner product between rendered normal maps (with approximate volumetric rendering for Gaussian splatting and NeRF representations) and the normal maps predicted from rendered RGB images:

$$R_{\text{normal}}(z) = \mathop{\mathbb{E}}_{c \sim \mathcal{C}} \left[ \left\langle h_{\text{normal}}(z, c), f_{\text{normal}}\big(h_{\text{RGB}}(z, c)\big) \right\rangle \right] \tag{10}$$

where $h_{\text{normal}}(z, c)$ and $h_{\text{RGB}}(z, c)$ denote the rendered normal map and rendered RGB images, respectively, from some 3D representation $z$ at camera pose $c$ in some camera pose set $\mathcal{C}$ and $f_{\text{normal}}$ is some pretrained normal map estimator, for which we use the one-step normal map prediction model in StableNormal [50] in our experiments.

Inspire by [53] which proposes to use depth-normal consistency to improve the geometric quality of reconstruction, we also propose to use the Depth-Normal Consistency (DNC) reward:

$$R_{\text{DNC}}(z) = \mathop{\mathbb{E}}_{c \sim \mathcal{C}} \left[ \left\langle h_{\text{normal}}(z, c), T\big(h_{\text{depth}}(z, c)\big) \right\rangle \right] \tag{11}$$

where the pseudo normal map $T\big(h_{\text{depth}}(z, c)\big)$ can be computed by taking finite difference of 3D coordinates computed from the rendered depth map.

## 5 Experiments

### 5.1 Base model, baseline, metrics and prompt dataset

**Base models.** We consider two state-of-the-art and open-sourced base models: DiffSplat [18], available in two variants finetuned from PixArt-$\Sigma$ [3] and StableDiffusion-v1.5 [37] (SD1.5 for short), and GaussianCube [54]. We use 20-step and 50-step first-order SDE-DPM-Solver [26] for DiffSplat-Pixart-$\Sigma$ and GaussianCube, respectively, and 20-step DDIM [39] for DiffSplat-StableDiffusion1.5. Unless otherwise specified, we use the DiffSplat-PixArt-$\Sigma$ [3] for experiments.

**Baseline methods.** With the 3D reward defined as an expectation of 2D rewards, other alignment methods can be similarly applied once we use stochastic samples to compute 3D rewards. Specifically, we consider these baseline methods: 1) DDPO [2], which finetunes diffusion models with the vanilla policy gradient method, 2) ReFL [48], which directly optimizes $R(\hat{z}_\theta(z_t))$ with a truncated computational graph $z_{t+1} \rightarrow z_t \rightarrow \hat{z}_\theta(z_t)$ and a randomly sampled $t$, and 3) DRaFT [4], which directly optimizes $R(z_0)$ with a truncated computational graph $z_K \rightarrow z_{K-1} \rightarrow ... \rightarrow z_0$. For ReFL, we sample $t$ from 15 to 19; for DRaFT, we use $K = 1$.

**Evaluation metric.** Following alignment studies in 2D domains [25, 60, 6], we consider three metrics: 1) average reward value, 2) multi-view FID score [10, 8] for measuring prior preservation and 3) multi-view CLIP similarity score [34] for measuring text-object alignment. The multi-view metrics for any given text prompt are computed by first computing metrics on each view and then taking the average over all views. Similarly, we compute metrics for each prompt and average them to obtain the final multi-view metrics. We use 60 unseen random prompts during the finetuning process for evaluation. For each prompt, we sample a batch of 3D assets (of size 32) to compute the metrics.

**Prompt dataset.** We use the prompt sets in G-Objaverse [32], a high-quality subset of the large 3D object dataset Objaverse [5]. For experiments on geometry rewards, we filter out the prompts for which the base models yield very low reward values.

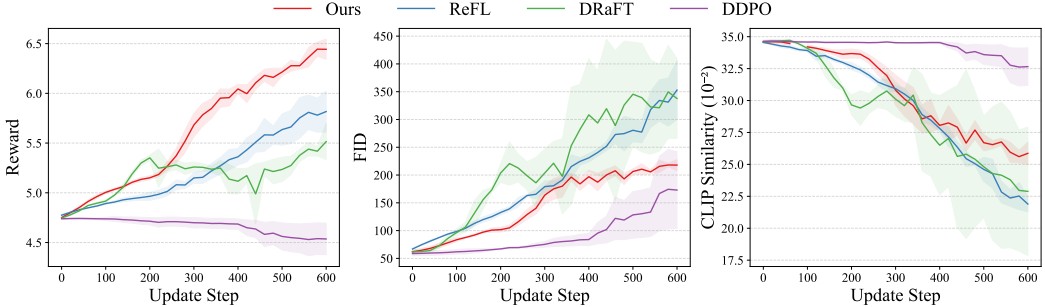

Figure 2: Convergence curves of metrics for different finetuning methods on Aesthetic Score. Our method achieves faster finetuning with better prior preservation and text-object alignment than ReFL and DRaFT. In addition, our method produces results with significantly lower variance in FID and CLIP similarity.

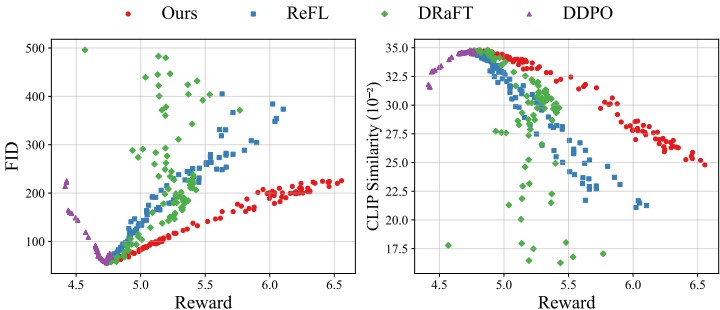

Figure 3: Trade-offs among reward maximization, prior preservation, and text-object alignment for various reward finetuning methods on Aesthetic Score experiments. Each data point represents evaluation results of model checkpoints saved at every 20 finetuning iterations. Models with higher rewards, higher CLIP similarity scores, and lower FID scores are considered superior. Our **Nabla-R2D3** shows better trade-offs between reward-improvement, and other metrics and consistently outperforms the baselines.

**Implementation Details.** We follow [25] and regularize the gradient updates: $\mathcal{L}_{\text{reg}} = \lambda \left\| \epsilon_\theta \left( x_t \right) - \epsilon_{\theta^\dagger} \left( x_t \right) \right\|^2$, where $\theta^\dagger$ is the diffusion model parameters in the previous update step and $\lambda$ is a positive scalar. We set $\lambda$ to $3e3$, $5e3$, $1e4$ for Aesthetic Score, HPSv2 and Geometry Reward respectively. During training, we sub-sample 40% of the transitions from each collected trajectory. For HPSv2 and Aesthetic Score experiments, we set the reward temperature $\beta$ to $2e6$ and $1e7$, respectively; for geometry rewards, we set $\beta$ to 1e6. To sample camera views $c$, we first sample four orthogonal views (front, left, back and right) with randomly sampled elevation $\pm 20°$ and then apply azimuthal perturbations by adding random offsets within a predefined range $\pm 60°$. We use a learning rate of $10^{-4}$ for ReFL, DDPO, DRaFT and **Nabla-R2D3**. The rest of the implementation details are elaborated in the appendix.

## 5.2 Results

**General experiments.** In Tab. 1, we show the metrics (average over 3 random runs) of models finetuned on different reward models with different finetuning methods. Our **Nabla-R2D3** is shown to be capable of achieving the best reward value at the fastest speed, and at the same time preserving the prior from the base model plus text-object alignment. We show in Fig. 2 the evolution of different metrics (both the mean value and the standard deviation) for different methods. As higher rewards inevitably lead to worse prior preservation and text-object alignment, we illustrate which method achieves the best trade-off by presenting the Pareto frontiers of all methods. Specifically, we plot the results from different checkpoints saved at various fine-tuning iterations of each independent run (with different random seeds) in Fig. 3. Furthermore, in Fig. 4, 5 and 6, we visualize the generated assets with the corresponding reward values from different finetuning methods and demonstrate that our method can qualitatively yield better alignment for various reward models meaningful for 3D generation. To illustrate that our method leads to more robust finetuning, we show in Fig. 7 the assets generated using the same random seeds with models finetuned with **Nabla-R2D3** and DRaFT. The shape with the DRaFT-finetuned model exhibits the severe Janus problem, where rendered multi-view

| Prompt | Egyptian cat head on a stone base. | A colorful coffee cup and saucer with a spoon, straw, and Swiss content design. | Green and gold Celtic treasure chest with intricate designs. | Mario doctor in a lab coat holding a pill. |

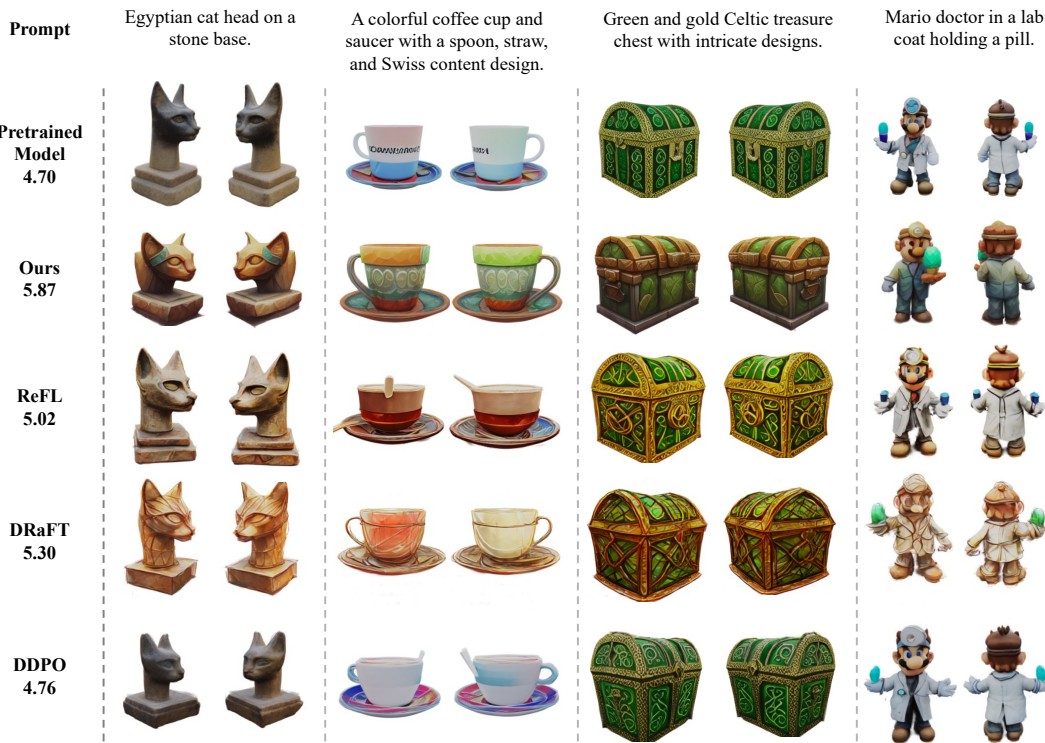

Figure 4: Qualitative comparison of 3D assets produced by models finetuned with different methods on Aesthetic Score. We show for each method on the left the average reward of the visualized assets. For fair comparison, we pick the model checkpoints that generate the highest rewards but without significant overfitting patterns.

images display inconsistent or contradictory characteristics from different viewpoints, while the one from **Nabla-R2D3**-finetuned model does not.

Table 1: Quantitative comparison between our method and the baselines on different reward models. Since it would be possible to trade FID with reward, we further present what the rewards are if FIDs are similar (with early stopping) in Tab. 5.

| Method | Aesthetic Score | | | HPSv2 | | | Normal Estimator | | |
|---|---|---|---|---|---|---|---|---|---|
| | Reward↑ | FID↓ | CLIP-Sim↑$(10^{-2})$ | Reward↑$(10^{-2})$ | FID↓ | CLIP-Sim↑$(10^{-2})$ | Reward↑$(10^{-2})$ | FID↓ | CLIP-Sim↑$(10^{-2})$ |
| Base Model | 4.72 | 55.26 | 34.58 | 22.86 | 55.26 | 34.58 | 89.48 | 55.26 | 34.58 |
| ReFL | 5.82 | 352.97 | 21.89 | 24.92 | 274.44 | 32.40 | 90.60 | 112.06 | 33.99 |
| DDPO | 4.54 | **172.95** | **32.64** | 22.23 | **69.59** | 34.35 | 89.45 | **63.93** | **34.56** |
| DRaFT | 5.51 | 337.77 | 22.89 | **32.65** | 224.64 | 33.99 | 91.11 | 296.23 | 28.36 |
| Ours | **6.44** | 217.89 | 25.86 | 27.85 | 131.38 | **35.35** | **92.03** | 104.45 | 34.18 |

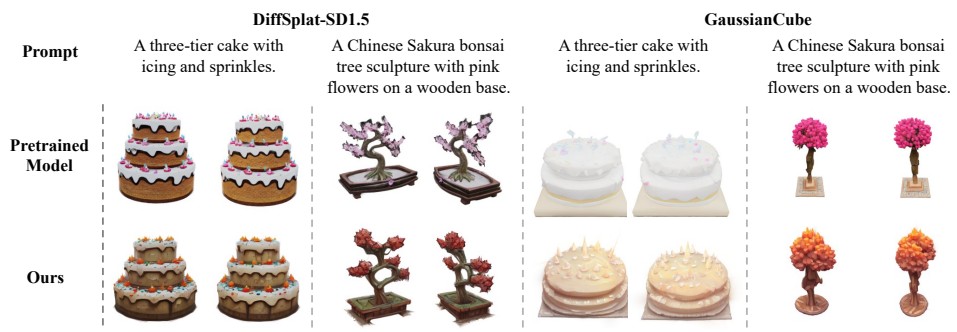

Figure 8: Finetuning results on two base models, DiffSplat-SD1.5 [18] and GaussianCube [54]. Our method generalizes well to models beyond DiffSpliat-PixArt-Σ.

**Different 3D-native generative models.** We experiment with different base models, including DiffSplat-SD1.5 [18] and GaussianCube [54], to show that our method is universally applicable. Our

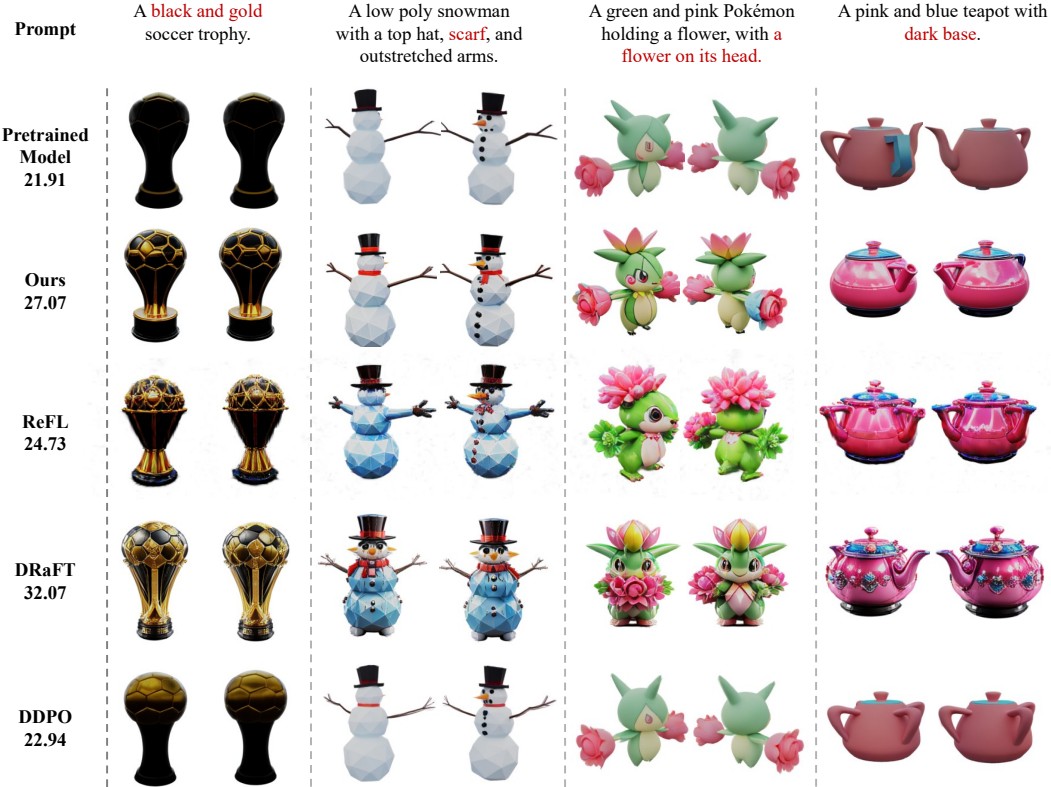

Figure 5: Quantitative comparison on HPSv2 [45]. For each object we present the front and back views. Prompts highlighted in red indicate unsuccessful instruction following by the base models. We further show the severe Janus problem of DRaFT in Fig. 7.

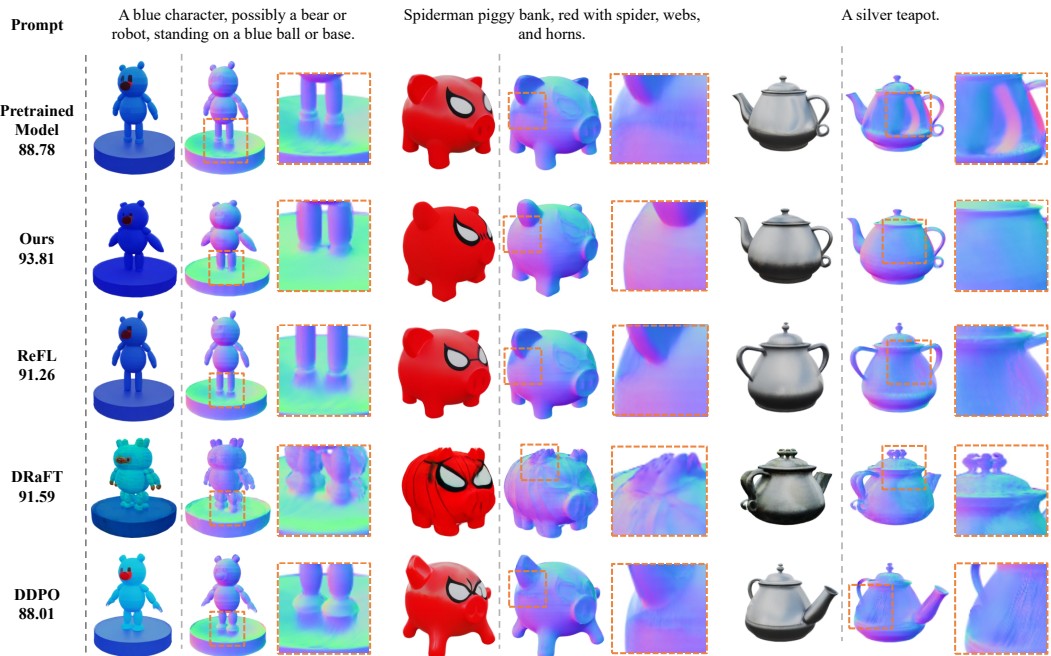

Figure 6: Quantitative comparison of different finetuning methods on the normal estimator reward (Eqn. 10). Left: front-view RGB image. Middle: front-view rendered normal map. Right: Zoomed-in details.

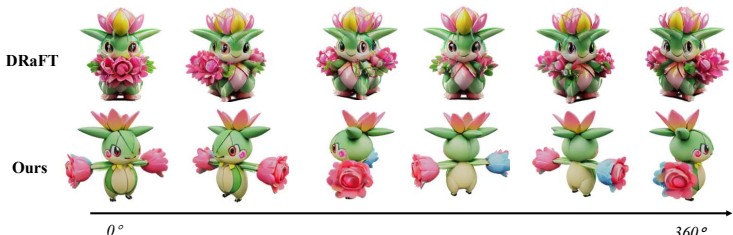

Figure 7: 360° visualization of 3D shapes generated by models finetuned with **Nabla-R2D3** and DRaFT. DRaFT-finetuned model is prone to overfitting and suffers from the Janus problem.

Table 2: Comparison of finetuning methods on different base models with the Aesthetic Score reward.

| Method | DiffSplat-SD1.5 | | | GaussianCube | | |
|---|---|---|---|---|---|---|
| | Reward ↑ | FID↓ | CLIP-Sim ↑$(10^{-2})$ | Reward ↑ | FID ↓ | CLIP-Sim ↑ $(10^{-2})$ |
| Base Model | 4.81 | 67.76 | 34.66 | 4.40 | 64.22 | 28.60 |
| ReFL | 6.08 | 396.79 | 20.66 | 5.50 | 296.46 | 17.31 |
| DDPO | 4.80 | **54.79** | **34.61** | 4.06 | 230.48 | 18.90 |
| DRaFT | **6.40** | 395.84 | 19.08 | **6.04** | 352.16 | 15.52 |
| Ours | 6.02 | 154.61 | 30.95 | 5.92 | **234.96** | **22.57** |

results (Fig. 8 and Tab. 2) show that our model consistently outperforms other finetuning methods and delivers desirable 3D assets.

**Comparison with 2D-SDS-based lifting alignment method.** We compare in Fig. 10 our method with DreamReward [51], a method that incorporates reward gradients into 2D-lifting-based 3D generation. Our method produces more visually-desirable shapes not only because 2D-lifting methods are less robust at synthesizing 3D shapes [31, 44] but also because the 3D-native generative model provides more 3D prior.

Table 3: Comparison with 3D-SDS.

| Method | Reward ↑ | FID ↓ | CLIP-Sim ↑ |
|---|---|---|---|
| 3D-SDS ($\eta = 3$) | 5.38 | 194.98 | 0.31 |
| 3D-SDS ($\eta = 1$) | 5.27 | **97.99** | **0.34** |
| Ours (200 steps) | 5.29 | 114.45 | 0.32 |
| Ours (600 steps) | **6.24** | 205.16 | 0.26 |

**Comparison with native 3D prior guided SDS baseline.** To underscore the benefit of inference with a finetuned 3D-native diffusion model compared to the SDS-based sampling approaches, we experiment with SDS on the DiffSplat base model with 3D diffusion loss plus the multi-view 2D reward model of Aesthetic Score: $\nabla L = \epsilon_{3D}(z_t, t) - \epsilon - \eta \nabla_{z_0} \log R_{3D}(z_0)$ with reward strength $\eta$. We observe (Fig. 9) that the SDS approach indeed yields worse appearance and geometry, even with better 3D prior from the base 3D-native diffusion model and increasing the strength $\eta$ does not improve results (Tab. 3). Moreover, running the 3D-SDS inference takes around 5 minutes, wherear inference with the finetuned model takes only around 8 seconds.

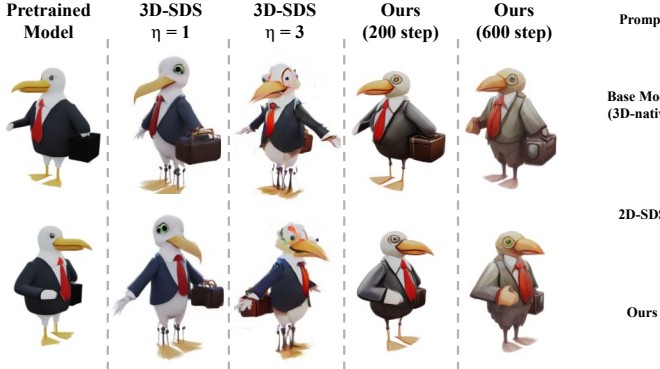

Figure 9: Comparison with 3D-SDS baselines on Aesthetic Score. We show two opposite views of the presented assets.

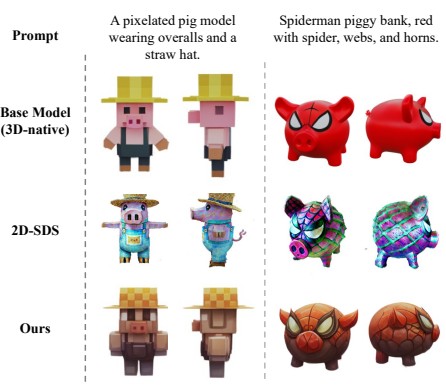

Figure 10: Comparison with 2D-SDS-based alignment method [51].

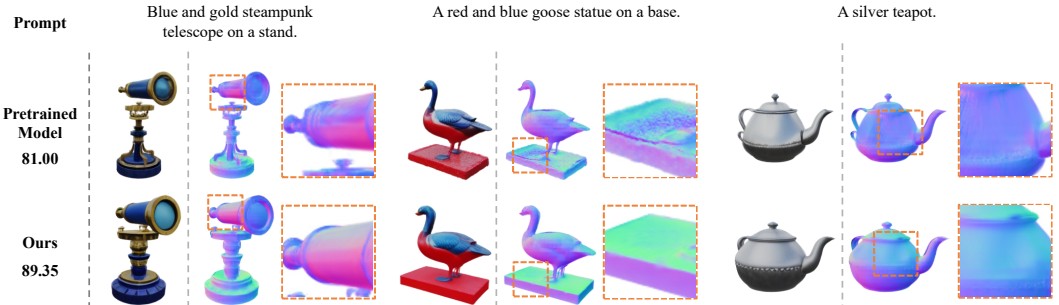

Figure 11: Qualitative comparison with pretrained model on DNC reward.

**Results on DNC reward.** We show the results of finetuning with different methods on DNC reward in Tab. 4. Our method achieves the best reward improvement without incurring much loss in FID and Clip-similarity. In Fig. 11, we visualize the geometry from the pretrained model and the model finetuned

Table 4: Comparison between the base model and the one finetuned with different methods on Depth-Normal Consistency.

| Method | Reward↑ ($10^{-2}$) | FID↓ | CLIP-Sim↑ ($10^{-2}$) |
|---|---|---|---|
| Base Model | 87.81 | 55.26 | 34.58 |
| ReFL | 89.15 | 158.19 | 32.74 |
| DDPO | 87.99 | **69.12** | 34.48 |
| DRaFT | 88.13 | 74.82 | **34.52** |
| Ours | **89.53** | 99.01 | 34.08 |

on the DNC reward. The results demonstrate that the DNC reward, independent of any priors from external normal estimators, can improve the sample geometry quality of 3D-native diffusion models. We further show the error map for the base model and the model finetuned with our method in Fig. 13.

**Visualization of the evolution process.** We visualize the evolution (every 50 update steps) of our method in the Aesthetic Score experiments (Fig. 12). We use the same seed, prompt, and initial noise, and we render the generated assets from the identical camera pose.

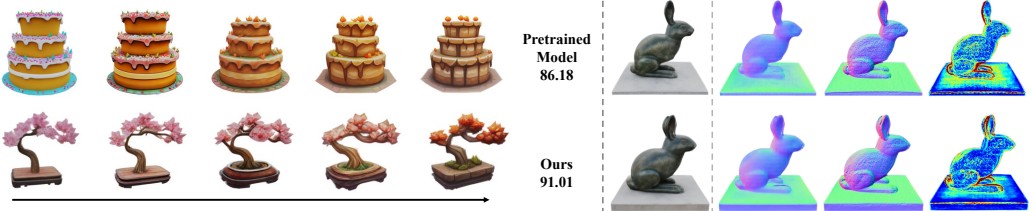

Figure 12: Visual evolution of the generated object with the same random seed during the finetuning process.

Figure 13: Results on DNC reward. From left to right: rendered RGB image, rendered normal map, depth-induced normal map, and the corresponding error map.

## 6 Discussions

**Reward finetuning vs. test-time scaling.** One may incorporate reward signals during inference using extra computational resources, a strategy known as test-time scaling [52]. Reward finetuning can be treated as an amortized process such that one does not have to pay the typically high cost of reward evaluation. Moreover, reward finetuning benefits from implicit regularization during network training, particularly in the case of LoRA finetuning [12].

**Limitations.** Our finetuning method suffers from the same issues as the lifting-from-2D approaches: no supervision for shape inner structures. Our method requires expensive gradient computations during the forward pass, underscoring the need for improved numerical algorithms and architectural designs for finetuning [33, 21]. Furthermore, our method focuses solely on parameter-level alignment and does not explore prompt-based alignment strategies [52].

## 7 Conclusion

We propose an efficient alignment method, dubbed **Nabla-R2D3**, for finetuning 3D-native generative methods with 2D differentiable rewards in a manner that avoids overfitting issues commonly seen in lifting-from-2D approaches. We demonstrate that **Nabla-R2D3** outperforms baseline methods across both appearance- and geometry-based reward models, as well as across different base architectures. The development of better alignment methods for diffusion models, including this work, contributes toward constructing virtual worlds aligned with human values.

## Acknowledgements

This project is supported by Key-Area Research and Development Program of Guangdong Province, China under Grant 2024B0101040004.

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

# Appendix

## Table of Contents

# A Overall algorithm

---

**Algorithm 1** 3D-Native Diffusion Alignment with 2D Rewards using **Nabla-R2D3**

---

1: **Inputs:** Pretrained diffusion model with sampling probability $p_{\text{base}}(x_t|x_{t+1})$, 2D reward model $R(\cdot)$
2: **Initialization:** Model to finetune with sampling probability $p_\theta(x_t|x_{t+1})$ where $\theta = \theta_{\text{base}}$.
3: Sample the initial batch of trajectories $\mathcal{D}_{\text{prev}} = \{(x_T, ..., x_0)_i\}_{i=1...N}$ with the current finetuned diffusion model $p_\theta$.
4: Set $\theta^\dagger \leftarrow \theta$.
5: **while** not converged **do**
6:    Sample a batch of trajectories $\mathcal{D}_{\text{curr}} = \{(z_T, ..., z_0)_i\}_{i=1...N}$ with the finetuned diffusion model.
7:    Subsample the time steps to train with: the full set $\mathcal{T}_i = \{0, ..., T-1\}$ or the sampled set $\mathcal{T}_i = \text{Sample-N}(\{0, ..., T-1\})$ where Sample-N is some unbiased sampling algorithm to randomly pick $N$ samples.
8:    Append the last timestep $t = 0$ to the sample trajectory set above.
9:    Sample a set of camera poses $c_1, ..., c_M$.
10:    Compute the loss ($f$ is the output of the diffusion denoising network)
$$\frac{1}{M(N+1)} \sum_{j=1}^M \sum_{t \in \mathcal{T}_i}$$
$$\left\| \nabla_{z_t} \log \tilde{p}_\theta(z_t|z_{t+1}) - w_t \beta \bar{\nabla} \left[ \nabla_{z_t} \log R(h(\hat{z}_\theta(z_t), c_j))) \right] \right\|^2$$
$$+ \lambda \| f_\theta(x_{t+1}) - f_{\theta^\dagger}(x_{t+1}) \|^2.$$
11:    Set $\theta^\dagger \leftarrow \theta$.
12:    Gradient update of $\theta$ with the loss function.
13:    Set $\mathcal{D}_{\text{prev}} \leftarrow \mathcal{D}_{\text{curr}}$.
14: **end while**
15: **return** finetuned model $f_\theta$.

---

# B    Proof of Unbiasedness of Nabla-R2D3

We start with the original Nabla-GFlowNet forward loss (Eqn. 2) for random variable $z_t$ but with the 3D reward specified in Eqn. 5:

$$L = \left\| \nabla_{z_{t-1}} \log \tilde{p}_\theta(z_{t-1}|z_t) - \gamma_t \beta \bar{\nabla} \Big[ \nabla_{z_{t-1}} \mathop{\mathbb{E}}_{c \sim \mathcal{C}} \log R(h(\hat{z}_\theta(z_{t-1}), c)) \Big] - g_\phi(z_{t-1}) \right\|^2 \quad (12)$$

The corresponding gradient is

$$
\begin{aligned}
\nabla_\theta L = {} & -2 \Big\langle \nabla_\theta \nabla_{z_{t-1}} \log \tilde{p}_\theta(z_{t-1}|z_t), \gamma_t \beta \bar{\nabla} \Big[ \nabla_{z_{t-1}} \mathop{\mathbb{E}}_{c \sim \mathcal{C}} \log R(h(\hat{z}_\theta(z_{t-1}), c)) \Big] - g_\phi(z_{t-1}) \Big\rangle \\
& \nabla_\theta \left\| \nabla_{z_{t-1}} \log \tilde{p}_\theta(z_{t-1}|z_t) \right\|^2 \\
= {} & -2 \mathop{\mathbb{E}}_{c \sim \mathcal{C}} \Big\langle \nabla_\theta \nabla_{z_{t-1}} \log \tilde{p}_\theta(z_{t-1}|z_t), \gamma_t \beta \bar{\nabla} \Big[ \nabla_{z_{t-1}} \log R(h(\hat{z}_\theta(z_{t-1}), c)) \Big] - g_\phi(z_{t-1}) \Big\rangle \\
& \nabla_\theta \left\| \nabla_{z_{t-1}} \log \tilde{p}_\theta(z_{t-1}|z_t) \right\|^2 \\
= {} & \nabla_\theta L_{\text{forward}}(z_{t-1:t})
\end{aligned}
\quad (13)
$$

which proves the unbiasedness of the proposed loss in Eqn. 7. The proof for the reverse loss is similar.

# C    Application to Flow Matching Models

As discussed in prior works [6, 25], the popular generative model of flow matching [19] that samples $x_1 = x(1)$ via $\dot{x} = v(x, t), x_0 = x(0) \sim \mathcal{N}(0, I)$ can be turned into an equivalent diffusion model (but with a non-linear noising process). The sampling (denoising) process of this equivalent diffusion model is:

$$\mathrm{d}X_t = \left( v(X_t, t) + \frac{\sigma(t)^2}{2\beta_t \left( \frac{\dot{\alpha}_t}{\alpha_t} \beta_t - \dot{\beta}_t \right)} \left( v(X_t, t) - \frac{\dot{\alpha}_t}{\alpha_t} X_t \right) \right) \mathrm{d}t + \sigma(t) \, \mathrm{d}B_t, \quad X_0 \sim \mathcal{N}(0, I).$$

$$(14)$$

where $\sigma(t)$ is an arbitrary diffusion term. We may therefore use **Nabla-R2D3** to obtain a finetuned diffusion model (and therefore the corresponding probability flow ODE [40]) from a pretrained flow matching model.

For the special case of rectified flows [22] with $x_0 \sim \mathcal{N}(0, I)$, the velocity field $v(x, t)$ and the probability flow $p(x, t)$ are related via the following formula ([62], Lemma 1):

$$\nabla \log p(x, t) = - \left[ \frac{1}{t} x + \frac{1-t}{t} v(x, t) \right]. \quad (15)$$

The corresponding reverse process of the equivalent diffusion model is therefore [20, 49]:

$$dx = \left[ v(x, t) + \frac{\sigma_t^2}{2t} \big( x + (1-t) v(x, t) \big) \right] + \sigma_t dw \quad (16)$$

# D   Implementation Details

**General experiments.** We use LoRA [12] parametrization with a rank of 16 on all attention layers plus the final output layer for DiffSplat-Pixart-$\Sigma$ [3], and a rank of 8 for DiffSplat-SD1.5 [18] and GaussianCube [54]. The CFG scales are set to 7.5, 7.5, and 3.5 for DiffSplat-Pixart-$\Sigma$, DiffSplat-SD1.5, and GaussianCube, respectively. All experiments were conducted with either two Nvidia Tesla V100 GPUs or GeForce GTX 3090 GPUs. It takes no more than one day to finetune models with our method and all other baselines.

**3D SDS.** During 3D SDS optimization, we sample timesteps $t \in [0, 400]$ and train the object for 1000 steps. For each step, the rewards are evaluated from four randomly sampled views. To compute test metrics, we sample 32 objects for each of 30 selected prompts, compute the mean per-prompt metrics and take the metrics averaged over all prompts.

# E   Comparison with Multi-view Generative Model

To emphasize the importance of 3D native representations for 3D consistency after finetuning, we use our method to finetune MVDream, a multi-view diffusion model, on Aesthetic score and compare the results with those with DiffSplat. The generated multi-view images are passed to Large Multi-view Gaussian Model (LGM) [41] to build reconstructed 3D objects so that they can be directly compared to 3D objects generated by 3D native models. Qualitative comparisons are presented in Fig. 14. The results from MVDream exhibit noticeable artifacts (illustrated in the green boxes in the figure), whereas the objects produced by the finetuned 3D native models do not.

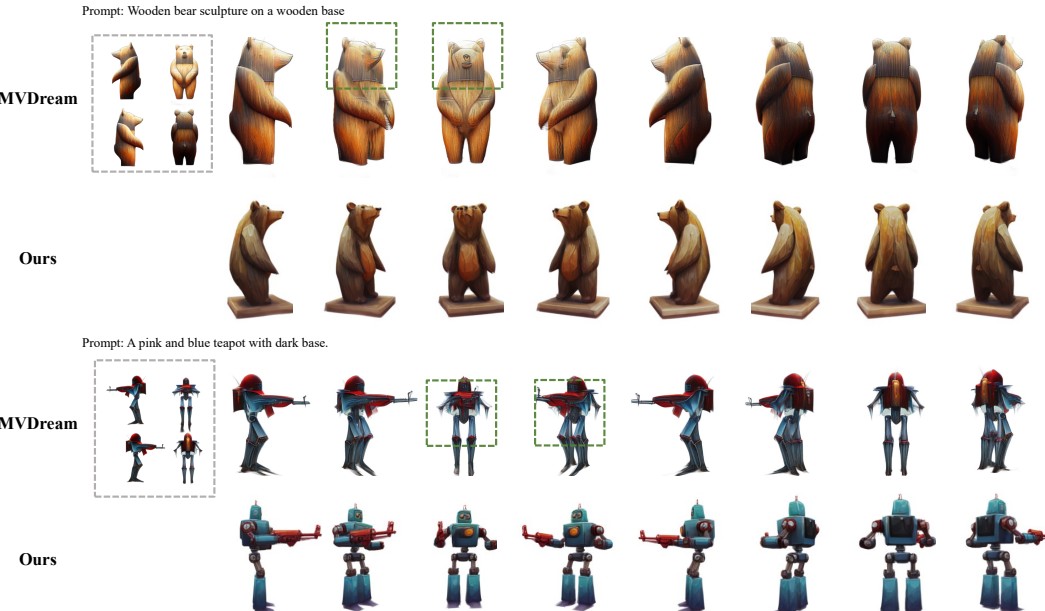

Figure 14: Qualitative comparison with MVDream. For MVDream, we show the multi-view images directly generated by the finetuned model in the gray dashed boxes; all other images for MVDream are rendered from the objects reconstructed by LGM (with the corresponding generated multi-view images).

# F    More Ablation Studies and Visualization

**Effect of different reward temperatures.** We experiment with different temperature values $\beta \in \{5e5, 1e6, 2e6\}$, and observe in Fig. 15 that a higher temperature leads faster convergence at the cost of worse text-object alignment and prior preservation. Since previous experiments (Fig. 2) showed that the variance of our method's metric is minimal, the statistics for the standard deviation (std) are omitted here.

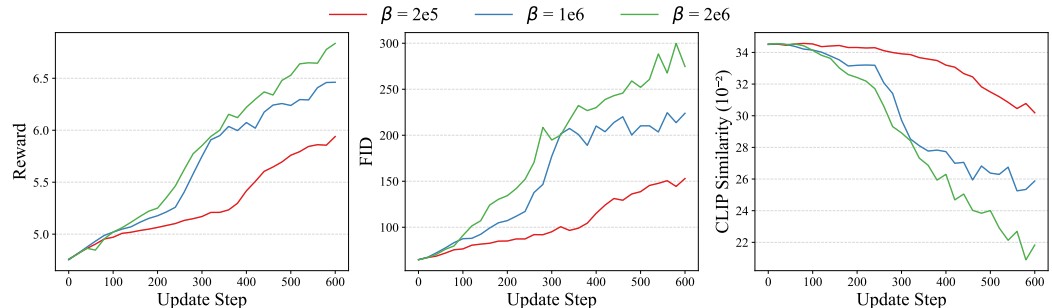

Figure 15: The relationship between the temperature parameter $\beta$ and Reward, FID scores, and CLIP-Similarity scores. Higher values of $\beta$ result in faster convergence, but at the cost of worse text-object alignment and diminished prior preservation.

**Effect of different learning rates.** As illustrated in Fig. 16, higher learning rates lead to faster convergence, but with compromises in prior preservation and text-object alignment.

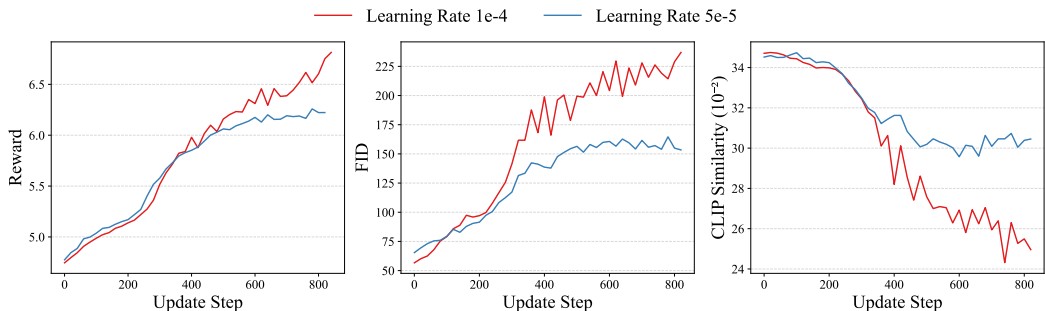

Figure 16: Convergence curves of metrics on different learning rate.

**Comparison of metrics given the same FID level on HPSv2 Reward.** We show the results on HPSv2 (with DiffSplat-Pixart-$\Sigma$) in Tab. 5, where we select checkpoints with roughly the same FID. Specifically, we pick checkpoints for models finetuned with ReFL and DRaFT such that their FIDs are approximately equal to that of our model in Tab. 1. The results clearly show that, given a similar FID level, our method achieves higher reward and CLIP-Sim scores.

Table 5: Results on HPSv2 (with DiffSplat-Pixart-$\Sigma$ ).

| Method | Reward $\uparrow (10^{-2})$ | CLIP-Sim$\uparrow (10^{-2})$ | FID $\downarrow$ |
|--------|-----------|------------|-----|
| ReFL   | 22.18     | 34.11      | 127 |
| DRaFT  | 25.65     | 34.41      | 143 |
| Ours   | 27.85     | 35.35      | 131 |

# G More Qualitative Results

360° videos of the generated objects can be found at our project website: https://nabla-R2D3.github.io.

**Pretrained Model**    **Ours**

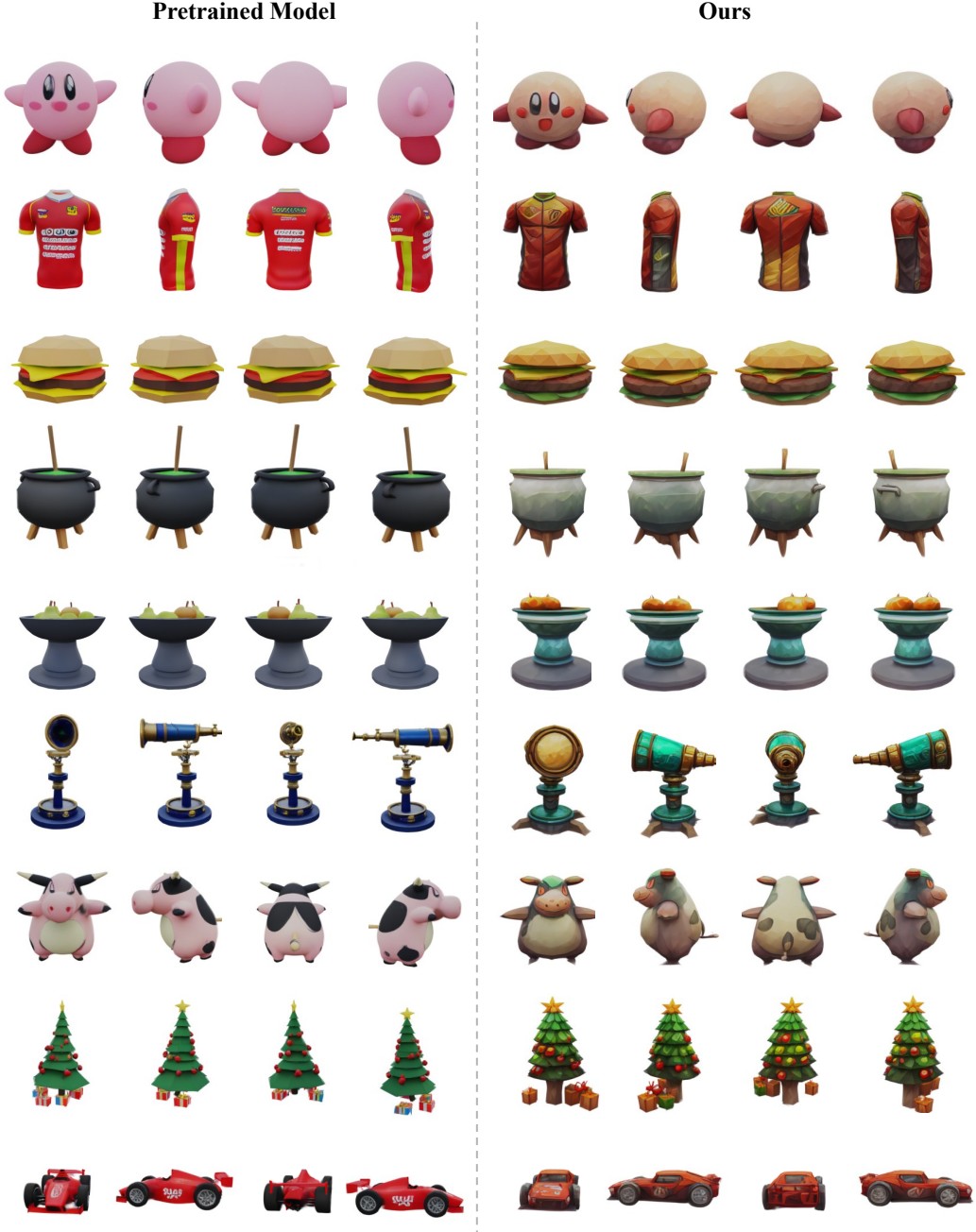

Figure 17: More qualitative results on Aesthetic Score.

**Pretrained Model**                    **Ours**

Prompt: A South Park cartoon character driving a green military jeep with a hat on it.

Prompt: A Mickey Mouse-themed candy apple, pixelated chocolate cake, and popsicle on sticks.

Prompt: Scientist character with green hair and hat, conducting experiments on a table with a plant and beaker.

Prompt: Yellow sphere character with a black hat and shoes.

Prompt: Purple and green toy robot on a purple base.

Prompt: A low poly a green dragon head with sharp teeth, yellow eyes, and spikes.

Prompt: Cartoon character wearing a pink hoodie with a triangle on it.

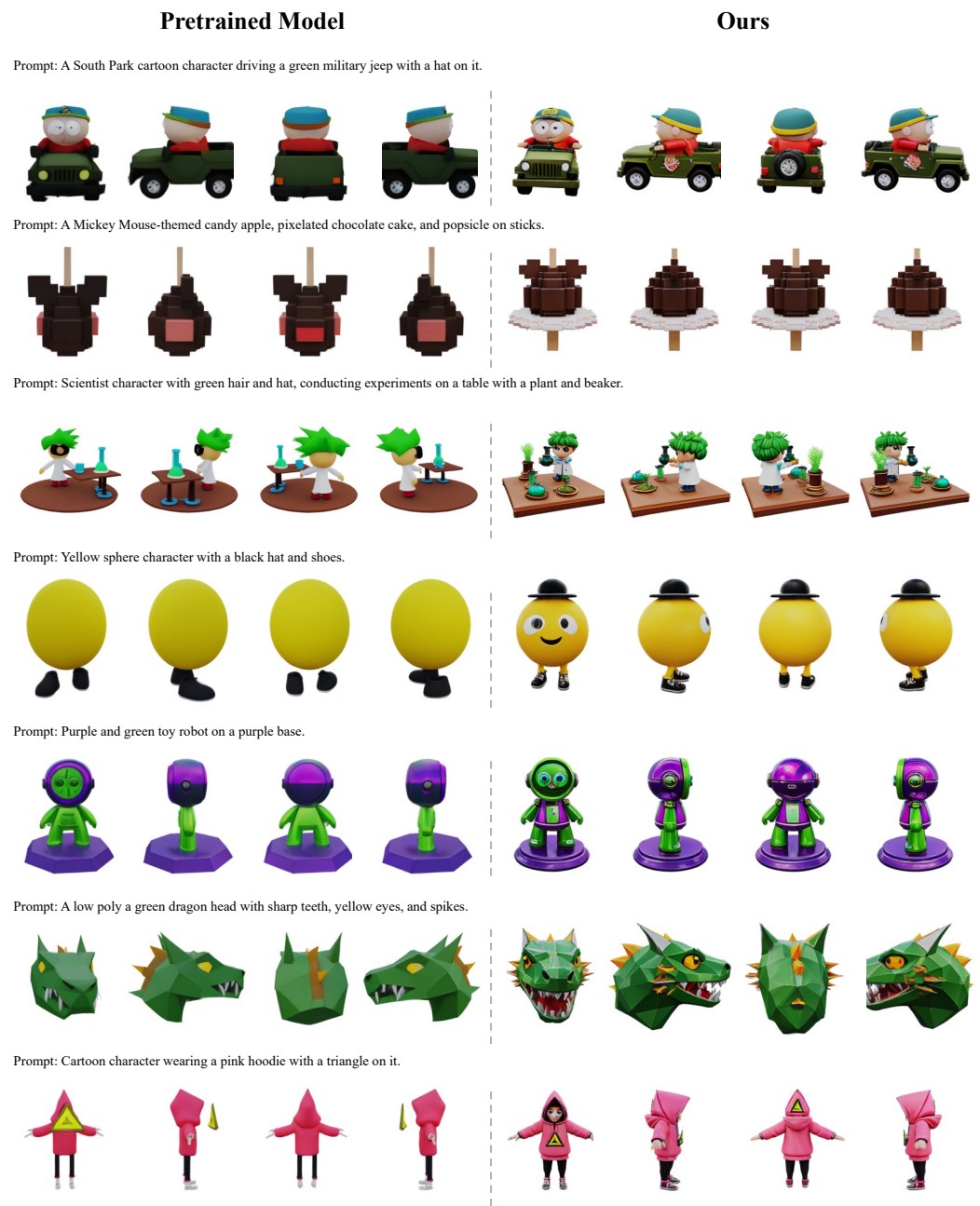

Figure 18: More qualitative results on HPSv2.

**Pretrained Model**

**Ours**

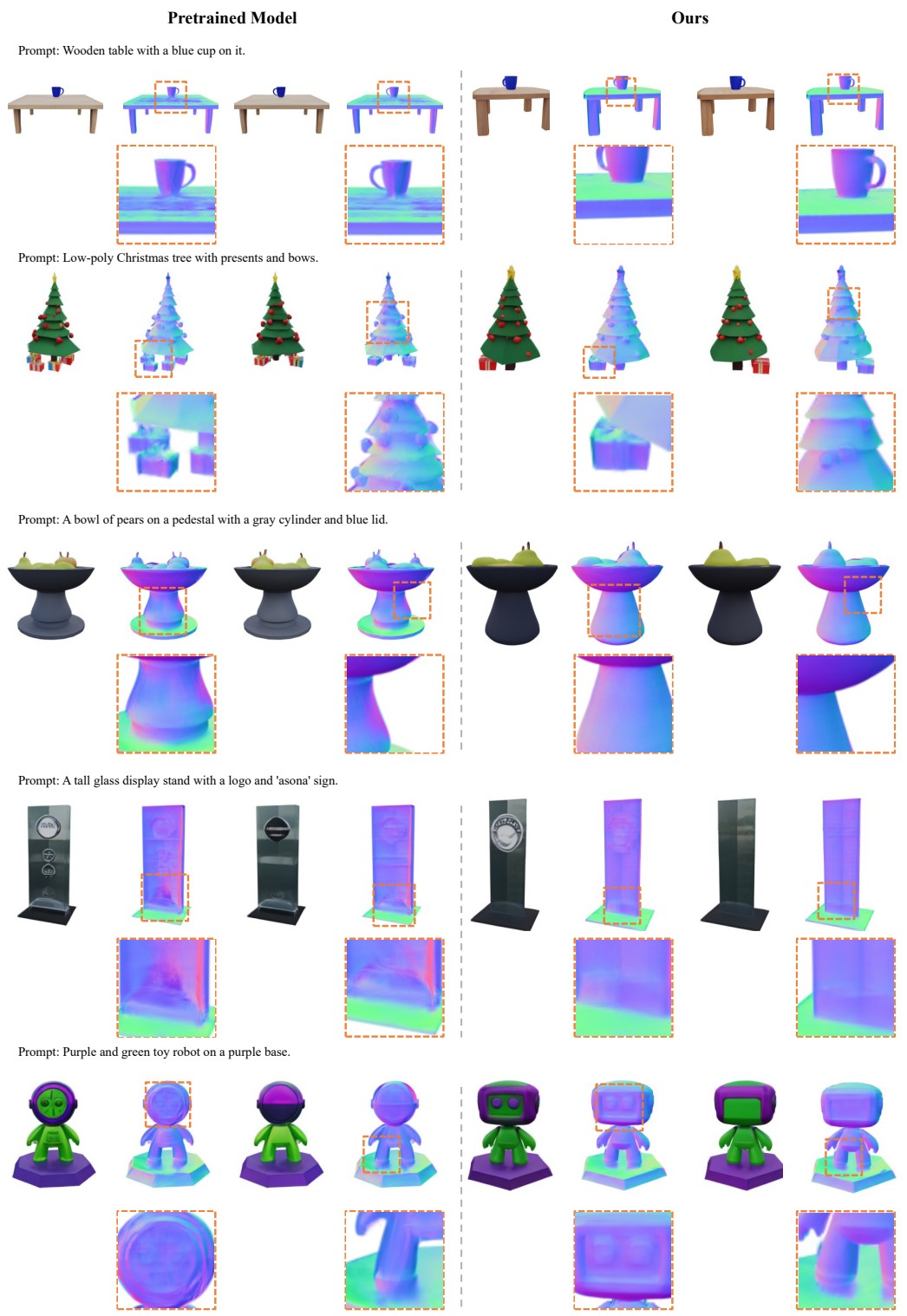

Figure 19: More qualitative results on the normal estimator reward.

# H Failure Cases

Similar to alignment for the 2D image domain, our method still suffers from two issues: 1) imperfect rewards and 2) reward hacking. The first one can easily be observed in our normal-estimator-based reward model, which inevitably will hallucinate non-existing geometry based on single-view RGB cues (Fig. 20). If we finetune the target diffusion model for too long, reward hacking becomes apparent as the generated shapes tend to over-optimize the rewards in the way that the natural geometry and semantics are gradually forgotten (Fig. 21).

Prompt: A Farbello Gold grape juice carton with grapes on it.

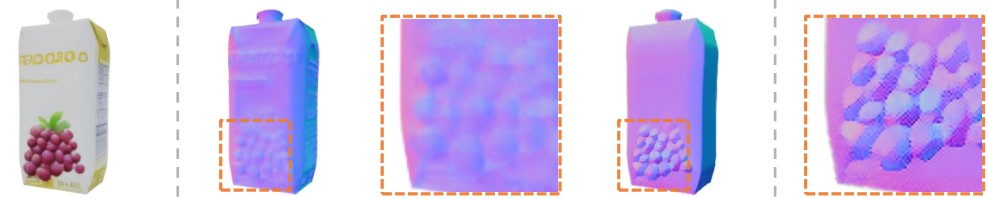

Prompt: Scientist character with green hair and hat, conducting experiments on a table with a plant and beaker.

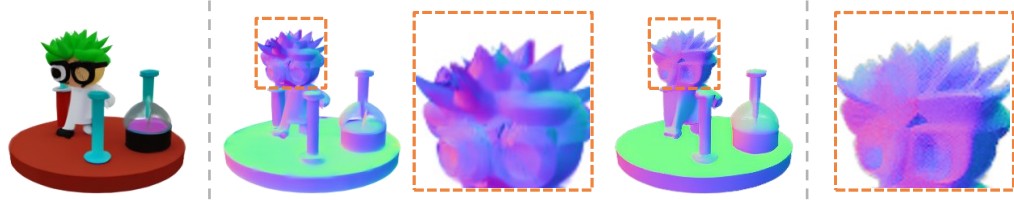

Figure 20: Failure cases of the normal estimator reward. Left: rendered RGB image, Middle: rendered normal map, Right: estimated normal map. The estimator produces wrong normal maps and therefore guides the diffusion model to generate wrong geometry.

Prompt: A green and blue striped toy llama.

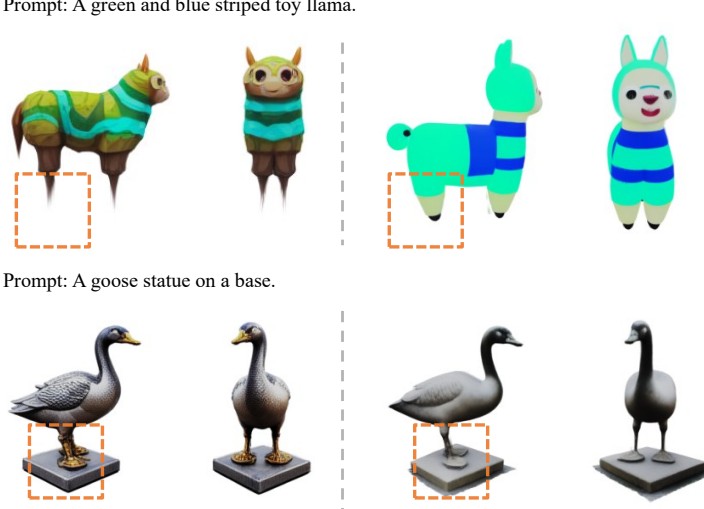

Prompt: A goose statue on a base.

Figure 21: Failure cases of the Aesthetic Score (first row) and HPSv2 (second row). The left column shows results from the finetuned model, while the right column presents results from the pretrained model. In both cases, the feet of animals become unnatural after finetuning.

