# OpenReview forum: "Nabla-R2D3: Effective and Efficient 3D Diffusion Alignment with 2D Rewards"
_NeurIPS.cc/2025/Conference — NeurIPS 2025 poster_

### Official Review · Reviewer_qR33 · 2025-07-01

**Clarity:** 2
**Significance:** 2
**Originality:** 2
**Rating:** 4
**Confidence:** 3

**Summary:**

This paper tackles the problem of improving a 3D generative model through RL-based fine-tuning with respect to 2D rewards. For this, the authors adopt a recently proposed Nabla-GFlowNet [22] framework and apply it to the 3D generation domain with some specific 2D rewards, e.g., appearance and geometry-based rewards. Experiments on 60 unseen prompts demonstrate that the proposed approach aligns the model better with the rewards (higher rewards values).

**Questions:**

1. My main concern is around the claim that
> (L53) Nabla-R2D3 can effectively, efficiently, and robustly finetune 3D-native generative models from 2D reward models with better preference alignment, better text-object alignment, and fewer geometric artifacts.

I do not find this convincing, especially with quantitative metrics in Tab. 1 and 2 as FID and CLIP-Sim seem quite inferior (in Tab. 2, the rewards are not good either). Further, I do not find the qualitative results in Figs 3, 4, 5, and 7 compelling either.

Even on the convergence speed, e.g., Fig. 2, my sense is that the baseline approaches, e.g., ReFL or DRaFT, perform better.

Can authors clarify?

2. About the novelty,
> (L48) to the best of our knowledge, the first method that effectively aligns 3D-native diffusion models with human preferences using only 2D reward models.

I am not fully convinced by this as it is a common practice to use 2D metrics to supervise 3D learning (on generations), e.g., DreamFusion [28], and a lot of works using LPIPS, a 2D perception metric, to provide learning signals.

Further, the framework largely mimics Nabla-GFlowNet [22] and does not present strong novelty when it comes to the methodology.

**Ethical Concerns:**

["NO or VERY MINOR ethics concerns only"]

**Final Justification:**

After reading the authors' rebuttal, some of my concerns have been addressed.

As discussed with the authors, though I agree that the findings are promising, I still think that, methodologically, it is straightforward. I think this paper should be positioned as a study paper, which requires clearer and careful refinement of the contributions. Thus, I raised my score to "borderline accept".

**Limitations:**

See the points in "questions".

**Quality:**

2

**Strengths And Weaknesses:**

1. Quality-wise: the qualitative results displayed in the paper looks good
2. Clarity-wise: the paper is generally well-written
3. Significance-wise: the paper tackles an important problem of how to align 3D generative assets with human preferences
4. Originality-wise: the paper demonstrates the potential for utilizing 2D rewards for 3D asset generation quality improvement

---

> ### Author Rebuttal · Authors · 2025-07-31
>
> We thank the reviewer for their time and efforts in reviewing our paper and raising their concerns over our claims.
>
> > I do not find this convincing, especially with quantitative metrics in Tab. 1 and 2 as FID and CLIP-Sim seem quite inferior (in Tab. 2, the rewards are not good either). Further, I do not find the qualitative results in Figs 3, 4, 5, and 7 compelling either.
>
> We would like to first emphasize that we are facing a **"multi-objective optimization problem"**: we would like the reward to be higher, but higher reward inevitably hurts instruction following and prior preservation. For this reason, we would argue against simply looking at the "bolded" figures in the table; instead, probably the best way to evaluate the methods is to fix certain metric to be some value and compare other metrics. Indeed, we **present the Pareto front in Figure 16 in the supplementary material**: we evaluate and visualize the **reward-FID and reward-CLIPScore pairs** for each saved checkpoint of finetuned models during the training process. On the reward-FID subplot, the lower-right corner is the best (high reward + low FID); on the reward-CLIPScore plot, the upper right corner is the best (high reward + high CLIPScore). In these two subplots, our method is clearly better than the other ones. We will bring this Figure 16 to the main text for better clarity.
>
> On Table 2: besides the "multi-objective optimization" issue, we would like to bring up the question: how effective one model is in finetuning models of different capacity? Just as finetuing larger LLMs is far more effective than finetuning small ones, in 3D generative models we have the same scaling law argument. We therefore argue that it is more important that one method works on large models.
>
> On qualitative results:
> - We would like to point the reviewer to Figure 6 in which we compare our method with DRaFT. The shapes generated by DRaFT-finetuned model shows the Janus problem (multi-head; all views look like the front view) while ours do not. We believe that this is a very solid evidence to prove that our method is very robust.
> - We also want to highlight some details in Fig 7: a) the baseline methods (ReFL and DRaFT) are more likely to produce "unaligned artifacts", for instance the 1st and 3rd instance on the DRaFT line; b) after finetuning, our model removes the incorrect estimation on the geometry (for instance, in the base model, specular reflections are mistakenly treated as the true geometry instead of surface material effects.
> - As the reward model of HPSv2 is trained with a preference dataset of text-image pairs, the results of Fig 4 should be evaluated in two dimensions: 1) aesthetics and 2) text-shape/texture alignment (or instruction following). For instance, the shape generated by the ReFL-finetuned model with the prompt "A low poly snowman with a top hat, scarf, and outstretched arms" does not have "scarf" in it.
>
> Besides, in the figures we purposely pick shapes with the highest rewards (averaged across views) yet without catastrophic forgetting of pretrained priors to demonstrate the advantage of our methods. Indeed, we can sacrifice the quantitative figures and pick shapes with lower rewards but better looking.
>
>
>
> > Even on the convergence speed, e.g., Fig. 2, my sense is that the baseline approaches, e.g., ReFL or DRaFT, perform better.
>
> Thanks for bringing this question up and we believe it demonstrates the key difference between 2D generation and "lifting-from-2D" generation. On 2D images, ReFL and DRaFT are indeed very fast. However, in lifting-from-2D, one samples a view and optimize the shape responsible for that view with limited 3D regularization. While one single view can be optimized at a fast speed, the optimization process may "oscillate" due to signals from different views. Here, stability and low gradient variance in training play greater roles in determining the convergence speed, which is why the reviewer observes the "counter-intuitive" results on convergence.
>
> > I am not fully convinced by this as it is a common practice to use 2D metrics to supervise 3D learning (on generations), e.g., DreamFusion [28], and a lot of works using LPIPS, a 2D perception metric, to provide learning signals.
>
> We respectfully disagree with the reviewer on this point. Here, we would like to first emphasize that our focuses are
>
> - to finetune a **3D-native** diffusion model with 2D rewards in a principled way, and
> - to show a set of plausible rewards for improving 3D-native diffusion models.
>
> We by no means ignore the predecessors in the direction of "lifting-from-2D" (e.g., DreamFusion as the reviewer mentioned, and many other earlier ones such as Dream Field [1] and GRAF [2]) and claim that we are the first to leverage 2D signals for 3D generation. Indeed, we have explicitly compared with SDS-based methods (i.e. DreamFusion) in Figure 9 and Table 3 and show that our method is better than these two alternatives.
>
> The key differences between out method and DreamFusion (or SDS-based methods in general) are that
>
> 1) SDS-based method aims to generate samples through optimization: initializing some random $x$ and use the diffusion loss $\mathbb{E} \|\| \epsilon_\theta(\text{noising}(x, t)) - \epsilon \|\|^2$ to iteratively update $x$. Our method finetunes a diffusion model and the samples are generated by conventional sampling procedures (DDPM, DDIM, DPM-solver, etc.) which starts from a Gaussian noise and follows a prescribed denoising schedule.
> 2) Our method is derived from a principled method with clear assumptions. By "principled", we mean that we know exactly what the error terms and how to learn the correction term. It is only that we for efficiency do not choose to learn this correction term. In DreamFusion, however, one does not have the theoretical guarantee that the optimization with the diffusion loss leads to the same distribution, both due to its sampling procedure with the SGD noise hard to characterize and due to that the diffusion loss is only a approximation of the ground-truth likelihood function (see Theorem 3 in [3]).
>
> > Further, the framework largely mimics Nabla-GFlowNet [22] and does not present strong novelty when it comes to the methodology.
>
> While it seems that we merely borrow an existing ML method, we argue that our contributions are non-trivial to the domain of 3D generation. In particular, we argue that
>
> 1) We are the first to introduce this idea of RL finetuning of 3D diffusion models with 2D rewards.
> 2) Empirically, all other methods do not work well: slow reward convergence, not robust, Janus problem, etc. Ours is the only one that is both efficient and robust. Given these results on baselines, we would like to argue that such a path to combine 3D-native model and 2D rewards may be overlooked without our attempt to use the recent method of Nabla-GFlowNet.
> 3) We empirically show that some rewards, especially the geometric ones, can be used to improve 3D-native diffusion models. This is also something not done in the 3D RL alignment setting.
>
> We would like also to point out that the field of 3D generation is still suffering from the lack of data. The largest publicly available dataset is Objaverse-XL, which contains 10m uncleaned 3D objects. People typically use only a subset of about 1 to 2 million shapes. Considering the dimensionality (3D vs. 2D) and the number of images we have in 2D domain (billions of images), there is still a huge gap. Our method aims to connect the 3D foundation models with the rich knowledge we have in 2D domain, just as DreamFusion does, but in a more direct approach.
>
> In addition, there are, to our knowledge, few papers that use RL for post-training in 3D large generative models (most people use DPO-based methods [4, 5]; the name of Mesh-RFT [5] is a bit over-claiming as they use DPO methods instead of online RL ones). By showing that RL finetuning can leads to good performance with simple reward models, we view our paper as a signal to encourage the people in the field of 3D generation to try different approaches (especially given that RLHF in general behaves better than DPO if carefully tuned in LLM research).
>
> ---
> We hope that our answers may resolve some of the concerns of the reviewer.
>
> Citations:
>
> [1] Zero-Shot Text-Guided Object Generation with Dream Fields. Jain et al. CVPR 2022. https://arxiv.org/abs/2112.01455
>
> [2] GRAF: Generative Radiance Fields for 3D-Aware Image Synthesis. Schwarz et al. CVPR 2020. https://arxiv.org/abs/2007.02442
>
> [3] Maximum Likelihood Training of Score-Based Diffusion Models. Song et al. NeurIPS 2021. https://arxiv.org/abs/2101.09258
>
> [4] Direct Preference Optimization: Your Language Model is Secretly a Reward Model. Rafailov et al. NeurIPS 2023. https://arxiv.org/abs/2305.18290
>
> [5] Mesh-RFT: Enhancing Mesh Generation via Fine-grained Reinforcement Fine-Tuning. Liu et al. https://arxiv.org/abs/2505.16761

---

> > ### Comment · Reviewer_qR33 · 2025-08-05
> >
> > I thank the authors for their time and effort in addressing my concerns, some of which have been answered.
> >
> > ## Fig. 16 vs Fig. 2
> >
> > Regarding Fig. 16 vs Fig. 2: are they essentially the same thing? In essence, is Fig. 16 the combination of three subfigures in Fig. 2? Further, I am not sure whether `evaluating and visualizing the reward-FID and reward-CLIPScore pairs for each saved checkpoint of finetuned models during the training process` can be called the Pareto frontier as the Pareto frontier is composed of a set of "optimal" solutions under various constraints. I doubt checkpoints saved during the fine-tuning process are optimal.
> >
> > ## Contributions statements
> >
> > Please refine the contributions statements from L48 to L56. During the rebuttal, the authors mentioned:
> >
> > > our focuses are to finetune a 3D-native diffusion model with 2D rewards in a principled way
> >
> > "The principled way" is proposed by Nabla-GFlowNet [22] instead of this paper.
> >
> > > our focuses are to show a set of plausible rewards for improving 3D-native diffusion models.
> >
> > I agree.
> >
> > > We are the first to introduce this idea of RL finetuning of 3D diffusion models with 2D rewards.
> >
> > I may not agree. There are several prior works, e.g., [a]
> >
> > [a] Xie et al., Carve3D: Improving Multi-view Reconstruction Consistency for Diffusion Models with RL Finetuning. CVPR 2024.
> >
> > In summary, my main point is that this work is a study that applies Nabla-GFlowNet [22] to the 3D domain and shows promising results to utilize RL fine-tuning.

---

> ### Author Response · Authors · 2025-08-05
>
> We thank the reviewer for their response.
>
> > In essence, is Fig. 16 the combination of three subfigures in Fig. 2?
> > I doubt checkpoints saved during the fine-tuning process are optimal.
>
> Yes, they are essentially the same thing except for different visualization, but Fig. 16 emphasizes the trade-offs between different metrics once the convergence speed is ignored.
>
> We use this plot mainly to answer these two questions
>
> *1. How to compare different methods if we have infinite amount of time for training?*
>
> *2. How to compare different methods if one is good at one metric and the other is good at the other?*
>
> From a practical perspective, one typically performs early stopping at some point during the training process. Therefore, given infinite amount of training time, a reasonable way to evaluate different methods is to see the trajectory of one method completely dominates another, and if not, what the trade-offs (in the sense of early stopping) can be. We believe that Fig. 16 better demonstrate the trade-off here and we observe a large margin of our method compared to other ones.
>
> We use the name Pareto front mainly to indicate that we are able to control the stopping time of training and show the trade-offs. We agree that the name can be a bit misleading and we wonder if there are suggestions for better naming.
>
> > I may not agree. There are several prior works, e.g., [a]
>
> Thank you for the correction and we apologize for this mistake here. We should have said 3D-native models--this is consistent with our claims in the paper, as with multi-view models we not able to arbitrarily render any view and multi-view images may not 3D-consistent with each other.
>
> In the meantime we thank the reviewer for pointing out this paper and we will cite this paper with proper discussion on the commons and differences.
>
> > Please refine the contributions statements from L48 to L56. During the rebuttal, the authors mentioned: "our focuses are to finetune a 3D-native diffusion model with 2D rewards in a principled way"
>
> We respectfully disagree with the reviewer. In our paper, we starts from the assumption of log-reward of 3D objects is the average of log-reward of 2D rewards (in different camera views), and we apply the method to reach the ideal objective. Then, we explicitly say that we perform the approximation on the gradient of the flow $F(\cdot)$ term. We believe this is a principled way to reach the final algorithm, as one may also reach the same algorithm but by just making analogies with and taking intuitions from DreamFusion or score matching.
>
> Nevertheless, we believe that we should rephrase our sentence so that we deliver a clearer message.
>
> > In summary, my main point is that this work is a study that applies Nabla-GFlowNet [22] to the 3D domain and shows promising results to utilize RL fine-tuning.
>
> We partially agree with the reviewer on this point. We believe other important components are
>
> - the setting of using 2D rewards
> - exploration of 2D reward models useful in the setting
> - phenomena we observed
>
> In particular, we surprisedly see remarkable differences between methods: even for reward convergence, finetuning with our method is faster. We argue that we have provided insights on this 3D setting and explored different possibilities of 3D-native diffusion models and reward models and we believe that, even our paper on the methodology side is a bit straightforward, our paper does provide enough contributions to the field of 3D generation.
>
> We hope that the reviewer may take these factors into consideration and re-evaluate our paper.

---

### Official Review · Reviewer_Xv7Y · 2025-07-01

**Clarity:** 2
**Significance:** 2
**Originality:** 2
**Rating:** 4
**Confidence:** 4

**Summary:**

This work aims to solve the fine-tuning of 3D generative models using 2D reward functions. It is based on the general framework of GFlowNet and instantiates it in the context of 3D generative models. The proposed framework uses a collection of multiple 2D rewards from prior works. The experiments show that the proposed method performs similarly to prior alignment fine-tuning methods in 3D generation.

**Questions:**

* Please clarify in the rebuttal the motivation of using the “approximated loss” (Eq. 9) and why it works better than the accurate loss.
* Please clarify the confusing mathematical terms as I mentioned in the weakness section.

**Ethical Concerns:**

["NO or VERY MINOR ethics concerns only"]

**Final Justification:**

After reading the rebuttal and having further discussions with the authors during the rebuttal stage, most of my concerns from the first round were addressed. Hence, I'm leaning positive for my final reviews. However, due to the limited technical novelty as mentioned in my review and other reviewers' comments, I wouldn't recommend stronger accept for this submission. My final score is therefore borderline accept.

**Limitations:**

While the authors did discuss some limitations of the proposed method, I think it would be helpful to also discuss why the quantitative study shows very little advantages of the proposed method against prior works.

**Paper Formatting Concerns:**

None.

**Quality:**

2

**Strengths And Weaknesses:**

- **Strengths**:
	- Even though the results presented by the author didn’t show promising improvement, I believe using 2D reward function to fine-tune 3D generative models is a promising direction to address the lack of good large-scale 3D data.

- **Weakness**:
	- **Incremental technical contribution with weak results**: Technically, this paper substitutes terms in the general framework of GFlowNet (Sec. 3.2) with specific instantiations for 3D object generation (Sec. 4.1). Additionally, the reward functions used in Sec. 4.2. are all commonly used objectives in prior works, as the authors already mentioned. Yet, a potentially valuable study of understanding the importance of each reward used is missing. Therefore, the technical novelty is limited. This would have been fine if such a simple and straightforward adaptation leads to surprisingly better results than before. However, from Tab. 1-3, the results are merely on-par with prior arts, if not worse in some cases. So, it’s hardly convincing that the proposed simple adaptation marks some progress.
   - **A critical design choice seems arbitrary**: Eq. 9 presents an “approximated loss” of the losses in Eq. 7 & 8. The only explanation of this design choice is a seemingly post hoc “assumption” that the hφ term is not needed. This is not a principled justification of a critical design choice. Is the hφ term hard to compute? If not, what is the motivation behind dropping it, except a random or even accidental experiment? Meanwhile, if it is true that the “approximated loss” works better than the accurate loss, does it indicate any flaws of the theoretical foundation? Much rigorous discussion can be added here to reveal many potentially interesting insights. Yet, they are missing in the submission.
	- **Confusing mathematical notations**: (1) Undefined term, hφ. This term is first used in L129 and then used a few times in L151-154. It’s part of the objective so it’s an important term in my opinion. However, there’s no formal definition of what it means and how to get this function. The authors should clarify this. (2) The use of gφ is also quite confusing. It’s first used in Eq. 2 & Eq. 3, again without clear definitions. Then, when I compare Eq. 7 & 8 with Eq. 2 & 3, it seems this gφ term becomes hφ, and there’s a different g() function defined as the rendering function in L145. The authors should make these notations more clear and accessible.

---

> ### Author Rebuttal · Authors · 2025-07-31
>
> We thank the reviewer for their time and efforts in reviewing our paper and raising their concerns over our claims.
>
> > Even though the results presented by the author didn’t show promising improvement
>
> > Therefore, the technical novelty is limited. This would have been fine if such a simple and straightforward adaptation leads to surprisingly better results than before. However, from Tab. 1-3, the results are merely on-par with prior arts, if not worse in some cases. So, it’s hardly convincing that the proposed simple adaptation marks some progress.
>
> We first respectfully disagree with the reviewer on this "weak improvement" point. We believe the misunderstanding comes from our miss in explaining how the methods should be evaluated and we elaborate it below:
>
> The metrics cannot be taken at face values because we are facing a **"multi-objective optimization problem"**: we would like the reward to be higher, but higher reward inevitably hurts instruction following and prior preservation. For this reason, we would argue against simply looking at the "bolded" figures in the table; instead, probably the best way to evaluate the methods is to fix certain metric to be some value and compare other metrics. Indeed, we **present the Pareto front in Figure 16 in the supplementary material**: we evaluate and visualize the **reward-FID and reward-CLIPScore pairs** for each saved checkpoint of finetuned models during the training process. On the reward-FID subplot, the lower-right corner is the best (high reward + low FID); on the reward-CLIPScore plot, the upper right corner is the best (high reward + high CLIPScore). In these two subplots, our method is clearly much better than the other ones. We will bring this Figure 16 to the main text for better clarity.
>
> On qualitative results:
>
> - We would like to point the reviewer to Figure 6 in which we compare our method with DRaFT. The shapes generated by DRaFT-finetuned model shows the Janus problem (multi-head; all views look like the front view) while ours do not. We believe that this is a very solid evidence to prove that our method is very robust.
> - We also want to highlight some details in Fig 7: a) the baseline methods (ReFL and DRaFT) are more likely to produce "unaligned artifacts", for instance the 1st and 3rd instance on the DRaFT line; b) after finetuning, our model removes the incorrect estimation on the geometry (for instance, in the base model, specular reflections are mistakenly treated as the true geometry instead of surface material effects.
> - As the reward model of HPSv2 is trained with a preference dataset of text-image pairs, the results of Fig 4 should be evaluated in two dimensions: 1) aesthetics and 2) text-shape/texture alignment (or instruction following). For instance, the shape generated by the ReFL-finetuned model with the prompt "A low poly snowman with a top hat, scarf, and outstretched arms" does not have "scarf" in it.
>
> Besides, in the figures we purposely pick shapes with the highest rewards (averaged across views) yet without catastrophic forgetting of pretrained priors to demonstrate the advantage of our methods. Indeed, we can sacrifice the quantitative figures and pick shapes with lower rewards but better looking.
>
>
> > Yet, a potentially valuable study of understanding the importance of each reward used is missing.
>
> Thank you for bringing this up. Our paper aims to show that these rewards are available and with our method we can effectively use them for 3D-native generation. We do not mix these reward models together and in each experiment we only use one. We do share the same thought with the reviewer that it is fruitful to explore a good mixture of reward models, but since our purpose is to show the task itself, we leave this to the future work.
>
> > A critical design choice seems arbitrary
>
> The design itself is not that arbitrary. Using this one step estimation for the "value function" (or basically leveraging the Tweedie's formula) is common in probabilistic modeling literature (see Equation 9 and 10 in [1]) and also used by the people working on applications (although without theoretical justification, e.g. [2]). In the Nabla-GFlowNet paper [3], the 2nd-line in Table 1 (with $w_B = 0$) shows that even if the correction term is not properly trained (due to the discard of some equations that making the correction term underdetermined), the performance is good enough. We follow the existing literature and use the best approximation we can do to accelerate training.
>
> > Please clarify in the rebuttal the motivation of using the “approximated loss” (Eq. 9) and why it works better than the accurate loss.
>
> We apologize for not stating it clearing in our main text, but we do **not** claim in the paper that it is a better loss in the sense of final performance. It is, however, a much more efficient one because we do not need to train the correction term with potentially involves 2nd-order gradients and large computational graphs.
>
> > 1) Undefined term, hφ. This term is first used in L129 and then used a few times in L151-154. It’s part of the objective so it’s an important term in my opinion. However, there’s no formal definition of what it means and how to get this function. (2) The use of gφ is also quite confusing. It’s first used in Eq. 2 & Eq. 3, again without clear definitions. Then, when I compare Eq. 7 & 8 with Eq. 2 & 3, it seems this gφ term becomes hφ, and there’s a different g() function defined as the rendering function in L145. The authors should make these notations more clear and accessible.
>
> We apologize for the unclear and undefined notations in the main text. $g_\phi$ is the residual term that corrects the educated guess on the "flow" (or "value function" in the RL sense if we turn this problem into a suitable Markov decision process with some well defined intermediate rewards), and $h_\phi$ should be $g_\phi$ (we accidentally use two distinct symbols for the same thing). We will revise the text in the next draft of the paper and explain the preliminaries & notations in a clearer way.
>
> ---
>
> Finally, we would like to underscore our contributions to the community of 3D generation that are not clear enough or not mentioned in the paper:
>
> - Empirically, all other methods do not work well: slow reward convergence, not robust, Janus problem, etc. Ours is the only one that is both efficient and robust. Given these results on baselines, we would like to argue that such a path to combine 3D-native model and 2D rewards may be overlooked without our attempt.
> - In addition, there are, to our knowledge, few papers that use RL for post-training in 3D large generative models (most people use DPO-based methods [4, 5]; the name of Mesh-RFT [5] is a bit over-claiming as they use DPO methods instead of online RL ones). By showing that RL finetuning can leads to good performance with simple reward models, we view our paper as a signal to encourage the people in the field of 3D generation to try different approaches (especially given that RLHF in general behaves better than DPO if carefully tuned in LLM research).
>
> We hope that our answers may resolve some of the concerns of the reviewer.
>
> ---
>
> [1] Practical and Asymptotically Exact Conditional Sampling in Diffusion Models. We et al. NeurIPS 2023. https://openreview.net/forum?id=eWKqr1zcRv&noteId=MlaVPZvkC6
>
> [2] Human Motion Diffusion Model. Tevet et al. ICLR 2023. https://arxiv.org/abs/2209.14916
>
> [3] Efficient Diversity-Preserving Diffusion Alignment via Gradient-Informed GFlowNets. Liu et al. ICLR 2025. https://arxiv.org/abs/2412.07775
>
> [4] Direct Preference Optimization: Your Language Model is Secretly a Reward Model. Rafailov et al. NeurIPS 2023. https://arxiv.org/abs/2305.18290
>
> [5] Mesh-RFT: Enhancing Mesh Generation via Fine-grained Reinforcement Fine-Tuning. Liu et al. https://arxiv.org/abs/2505.16761

---

> ### Comment · Reviewer_Xv7Y · 2025-08-05
>
> I thank the authors for the thorough rebuttal to address my concerns. Some of my questions are properly addressed:
> - The justification of using the approximated loss in Eq. 9.
> - The confusion notations of hφ and gφ.
>
> Although I don't think patching up such important technical expositions after submission should be encouraged, I acknowledge that they can be fixed in final refinement of the draft.
>
> However, I may respectfully disagree with other arguments in the rebuttal:
>
> > The metrics cannot be taken at face values because we are facing a "multi-objective optimization problem": we would like the reward to be higher, but higher reward inevitably hurts instruction following and prior preservation.
>
> I think this argument has 2 flaws. First, `we would like the reward to be higher, but higher reward inevitably hurts instruction following and prior preservation.` doesn't justify for having lower instruction following and prior preservation. As we probably all agree, these 2 are important properties for any fine-tuned text-to-3D generative models. Thus, sacrificing them for other properties doesn't necessarily mean "overall improvement". Second, from the presented Tab. 1 and Tab. 2, even the rewards of Nabla-R2D3 are not clearly better. For instance, DRaFT sometimes clearly outperforms Nabla-R2D3 (e.g. HPSv2 in Tab.1). So overall, I'd read the reward-wise performance of Nabla-R2D3 on par with DRaFT.
>
> > instead, probably the best way to evaluate the methods is to fix certain metric to be some value and compare other metrics. Indeed, we present the Pareto front in Figure 16 in the supplementary material: we evaluate and visualize the reward-FID and reward-CLIPScore pairs for each saved checkpoint of finetuned models during the training process
>
> I find this figure hardly convincing, and unfortunately rather confusing. Many questions rise when I try to read this figure. Among them, the important ones are:
> - The most important question is, since ultimate the user can only pick one sample point for each method in Fig. 16, i.e. one model checkpoint, which one should the user pick and why? As the caption already mentioned, the main message from Fig. 16 is "trade-offs". That is, for almost all methods, higher reward scores lead to worse FID or worse text-alignment. One would then naturally ask should we seek higher rewards which inevitably push the FID to 100+ across the board and is know to have poor quality, similarly for text-alignment? Or the actual useful part of the plot is the bottom-left corner of the reward-FID plot where we have reasonable FID to maintain the quality and okay-ish reward scores? And in that corner, consistent with the messages in Tab. 1 and Tab. 2, there seems no clear difference among different methods.
> - How should I compare this figure with Tab. 1 or Tab. 2? What reward model is used in Fig. 16? Where can I find the data points of Tab. 1 and / or Tab. 2 in Fig. 16? In fact, I try to locate the DRaFT points but couldn't. I can roughly find the Nabla point but it is around the bottom left corner of Reward-FID and top left corner of RewardCLIP. This aligns with my guess in the last point that the actually valid area of Fig. 16 is actually those sub-regions.
> - Since the points are evaluated from different checkpoints of a training run, why not order them by iterations and connect the points, so that we can see how the metrics evolve over training? Scattering different checkpoints over training seems like a bad idea to me.
> - Why for DRaFT, there are multiple points near a vertical reward score line?
>
> The lack of clear performance improvement can be accepted if the paper explores a very novel and potentially very impactful idea. However, as reviewer qR33 pointed out, the technical novelty is limited to appling Nabla-GFlowNet [22] to the 3D domain. So, this doesn't constitutes a case of pioneering a brand new idea.
>
> In summary, while I appreciate the authors' efforts in clarifying my confusion in the rebuttal. I'm still leaning towards rejection unless the authors have other stronger arguments for the above concerns or if I misread some of the arguments and figures.

---

> ### Author Response · Authors · 2025-08-05
>
> We thank the reviewer for their response and their efforts in reading our long comments.
>
> > Thus, sacrificing them for other properties doesn't necessarily mean "overall improvement".
>
> > which one should the user pick and why?
>
> We do not mean that we intentionally sacrifice one metric for the other, but mean that the loss of prior preservation is inevitable as shown by the finetuning objective itself -- deviating from $p_\text{base}$ is a loss of prior info. In such an unfortunate case, how should how should we pick methods? For instance, if we have a FID budget, say that we aim for FID lower than 700, what is the maximum reward we can get from a reward model? This is why we try to emphasize trade-offs: **a better model should provide a better set of metrics pairs for users to pick**, and this is the "overall improvement" we aim for.
>
> This problem of picking checkpoints is inevitably empirical. In RLHF or DPO for LLMs, the $\beta$ value that controls how much to follow the reward is tuned with empirical ways; in classifier-free guidance for diffusion sampling, we set the guidance scale to some magic number like $5.0$ or $7.5$ as people find these values lead to a typically acceptable balance between instruction following, naturalness of images and sample diversity. Our setting is not an exception.
>
> We would also like to argue that the base model of DiffSplat is still not that powerful and one will find the trade-offs not satisfactory. With larger models trained with larger (and probably proprietary) datasets, one will observe better results in the sense that the rewards get higher without significant loss in FID. This is why we show the results on smaller models like DiffSplat-SDv1.5 and GaussianCube in Table 2.
>
> We further underscore that **one should not only look at reward values** due to reward hacking issues. Even in 2D domain, the problem of reward hacking is severe. For instance, in Fig 30 in the DRaFT paper, the authors show that when finetuning with DRaFT using Aesthetic Score, models with an average reward value of around 6 already produce visually unaccetable images (even if they have tried various regularization methods).
>
> > What reward model is used in Fig. 16? How should I compare this figure with Tab. 1 or Tab. 2?
>
> Sorry for not clearly explaining this experiment setting and the visualization method for Fig. 16. The reward model is Aesthetic Score and we use DiffSplat-PixArt-Σ as the base model, basically the same setting for the leftmost sub-table of Table 1 (let's call this subtable Table 1.1).
>
> The values in Table 1.1 are produced by
>
> - training models with 3 random seeds,
> - evaluating all models finetuned with 600 update steps, and
> - taking the average of results of all 3 random runs for each method
>
> Fig. 16 plots all pairs of metrics of all checkpoints (all random runs included) without any averaging. That is why Table 1 does not seem to correspond to Fig. 16. Instead, Fig. 16 should be compared with Fig. 2 where both the mean curves and the confidence intervals (the standard deviation) are visualized.
>
> > Why for DRaFT, there are multiple points near a vertical reward score line?
>
> It is shown in Fig. 2 that DRaFT is highly unstable during optimization. Plus that we visualize multiple random runs.
>
> > DRaFT sometimes clearly outperforms Nabla-R2D3 (e.g. HPSv2 in Tab.1)
>
> As we have argued above, if we are given sufficient amount of time, we should look at the trade-offs since one can stop training at any point. Unfortunately, for presenting tables, we have to pick a consistent way for showing the numerical values: this is why we end up showing results of the models all finetuned with 600 steps.
>
> To illustrate why the values for DRaFT for HPSv2 in Table 1 are not satisfactory, we show in Fig.6 that how DRaFT may lead to severely damaged objects (using the same prompt as shown in Fig. 4). Note that in Fig. 4 we pick early checkpoints of ReFL and DRaFT that do not lead to quality collapse so that the visual comparison is fair.
>
> We agree that trade-offs of metrics for HPSv2 are not shown clearly, for which we are re-running experiments and will get back to the reviewer shortly (to show reward + CLIPScore with roughly the same FID).
>
> > why not order them by iterations and connect the points, so that we can see how the metrics evolve over training? Scattering different checkpoints over training seems like a bad idea to me.
>
> Fig. 2 in our paper shows the evolution of all metrics over time. This scatter plot Fig.16 is supplementary and is mainly to emphasize the trade-off perspective (ignoring the convergence speed).
>
> > the technical novelty is limited to appling Nabla-GFlowNet to the 3D domain
>
> From a pure ML perspective, yes. But we still would like to argue from the perspective of the community of 3D generation -- it is important to leverage 2D data for post-training in 3D-native generation (especially considering the data scarcity issue in 3D), a direction that is still not fully explored.

---

> > ### Author Response · Authors · 2025-08-06
> >
> > Below is the table for HPSv2 (with DiffSplat-Pixart-Sigma) if we pick checkpoints with FIDs roughly the same. Specifically, we picked checkpoints for models finetuned with ReFL and DRaFT such that the FID is roughly the same as the FID of ours in Table 1.
> >
> > | Method       | Reward | CLIPScore | FID |
> > |--------------|----------|----------|----------|
> > | ReFL     | 22.18    | 34.11     | 127 |
> > | DRaFT   | 25.65    | 34.41      | 143 |
> > | Ours   | 27.85    | 35.35      | 131 |

---

> ### Comment · Reviewer_Xv7Y · 2025-08-08
>
> I thank the authors for the further clarification and the new results. I have a better idea of how to read the results. Based on the discussion here, I'd strongly suggest that the authors move the study of Fig. 16 to the main paper and add more details as described here to clearly describe how it is plotted. I'm happy to raise my final score to borderline accept based on our discussion here.

---

> > ### Author Response · Authors · 2025-08-08
> >
> > We are very glad that most of the reviewer's concerns are addressed and we sincerely appreciate the reviewer's engagement in the discussion. As suggested by the reviewer, we will definitely move Fig.16 to the main paper (and probably including an augmented version of this new table if space permits) and make sure that the explanations on settings and implications are clear to readers.

---

### Official Review · Reviewer_qnSr · 2025-07-03

**Clarity:** 3
**Significance:** 3
**Originality:** 2
**Rating:** 5
**Confidence:** 4

**Summary:**

The paper proposes to improve 3D generation from 2D diffusion by RLHF. Here the reward is not from human, but from models judging the aesthetics and geometry quality, applied on 2D rendered views.

**Questions:**

None.

**Ethical Concerns:**

["NO or VERY MINOR ethics concerns only"]

**Final Justification:**

The method is straight-forward and effective. I keep my original positive rating.

**Limitations:**

Yes.

**Paper Formatting Concerns:**

None.

**Quality:**

3

**Strengths And Weaknesses:**

**Strengths:**
1. The idea is a straight-forward adaptation of RLHF for LLMs to 3D generation. It is relatively easy to understand and simple to implement.
2. The reward is obtained by looking at the 2D rendered views instead of directly on the 3D asset, allowing us to use much more powerful reward models trained on 2D data.
3. Qualitative results (Figure 3, 4, 7) show that the method effectively improves the texture and geometry quality.

**Weaknesses:**
1. The idea is so straight-forward that novelty might be a concern. It is not a problem for me though. Proving that this idea can work in the 3D generation setting with strong evidence is good enough for me.

---

> ### Author Rebuttal · Authors · 2025-07-31
>
> We greatly appreciate the reviewer for acknowledging our contributions. Partially to respond to other reviewers as well, here we would like to emphasize that although the method is straightforward, all other variants do not work well, including:
>
> - Vanilla finetuning methods (ReFL, DRaFT, DDPO)
> - SDS-based generation with 2D base model + 2D reward
> - SDS-based generation with 3D-native base model + 2D reward
>
> Given these results on baselines, we would like to argue that such a path to combine 3D-native model and 2D rewards may be overlooked without our attempt to use the recent method of Nabla-GFlowNet.

---

### Official Review · Reviewer_2qff · 2025-07-03

**Clarity:** 3
**Significance:** 2
**Originality:** 3
**Rating:** 4
**Confidence:** 3

**Summary:**

This paper proposes a novel finetuning method for 3D generative diffusion models that relies solely on pretrained 2D models to provide reward signals. The method enables efficient optimization of 3D generation quality, resulting in models with richer geometric details and improved stability. The core of the approach is a gradient-based reward mechanism: by rendering multi-view images from 3D shapes and scoring them using a pretrained 2D model, reward gradients with respect to the latent variables are computed and used to guide the corresponding gradients on the side of the 3D diffusion model for optimization. In addition, optimization constraints are applied to both ends of the diffusion trajectory between adjacent latent variables to improve training stability. The method demonstrates significantly higher rewards and mitigates prior forgetting, even with a limited number of finetuning steps.

**Questions:**

1. It seems that the model doesn't respond well to color-related prompts. Does it show similar insensitivity to other types of prompts as well?
2. The generated assets appear to share a relatively uniform style. Is the model capable of producing assets with diverse styles?

**Ethical Concerns:**

["NO or VERY MINOR ethics concerns only"]

**Final Justification:**

The rebuttal has addressed most of my concerns. I will keep my rating.

**Limitations:**

yes

**Quality:**

3

**Strengths And Weaknesses:**

Strengths:

1. Compared to other finetuning methods, this approach generally produces results with better 3D consistency across views.

2. Generated textures typically show higher quality and clearer details relative to alternative finetuning techniques.

3. The pipeline achieves fast finetuning convergence and low inference latency.

Weaknesses:

1. The alignment between generated results and input prompts is not always consistent. For example, as demonstrated in the paper (e.g., variation in flower color), some prompt-driven attributes are not accurately retained.

2. There can be noticeable deviations in visual style compared to the baseline model.

3. The overall performance, including prompt fidelity and visual quality, largely depends on the quality of the pre-trained 2D backbone.

---

> ### Author Rebuttal · Authors · 2025-07-31
>
> We thank the reviewer for their time and efforts in the reviewing process and their acknowledgement of our contributions.
>
> > The alignment between generated results and input prompts is not always consistent. For example, as demonstrated in the paper (e.g., variation in flower color), some prompt-driven attributes are not accurately retained.
> > There can be noticeable deviations in visual style compared to the baseline model.
>
>
> This behavior depends on two factors: 1) how much we would like the model to follow the reward instead of the pretrained prior and 2) what outcomes the reward model is rewarding. For instance, the reward model of Aesthetic Score favors image aesthetics and does not take any text prompt as input. Therefore, with a high value of $\beta$ (the parameter to control the strength of the reward), the generated shapes inevitably deviate from the prompt. Such a problem can be alleviated by carefully picking the $\beta$ parameter. We may also consider combining different reward models, which we leave for future drafts of this paper or future work.
>
> > The overall performance, including prompt fidelity and visual quality, largely depends on the quality of the pre-trained 2D backbone.
> >
>
> Yes, this is another inevitable fact: just as finetuning larger LLMs is more effective than finetuning smaller ones (examples can be found in RL finetuning of LLMs for math reasoning), if the base model is largely incapable of generating reasonable geometries and textures, we cannot expect finetuning to greatly improve it.
>
> > It seems that the model doesn't respond well to color-related prompts. Does it show similar insensitivity to other types of prompts as well?
>
> Yes, we did observe insensitivity to some other types of prompts:
>
> - Text-generation-related (e.g., generate the word "Apple" as a 3D shape)
> - Small objects (e.g., a spoon on the tea cup)
> - Directions (left or right)
>
> We believe that these issues are mostly due to the incapability of the base models. In the meanwhile, it possibly reveals that we need better reward models, especially dense rewards (e.g., [1]).
>
> > The generated assets appear to share a relatively uniform style. Is the model capable of producing assets with diverse styles?
>
> Thanks for bringing this question up. We also notice that after finetuning for too long, the styles tend to converge. The reason behind this behavior, we believe, is hinted by your questions above: the base model is still not that capable, plus that we use a relatively large $\beta$ value. We do observe that if we do early stopping, the styles are far more diverse, which partially verifies the claim.
>
> ---
>
> [1] Rich Human Feedback for Text-to-Image Generation. Liang et al. CVPR 2024. https://arxiv.org/abs/2312.10240

---

> > ### Comment · Reviewer_2qff · 2025-08-04
> > **follow-up comment**
> >
> > Thanks for the response. The rebuttal has addressed most of my concerns. I will keep my rating at this time.

---

> > > ### Author Response · Authors · 2025-08-05
> > >
> > > We are glad that most of the reviewer's concerns are addressed and we appreciate the reviewer's acknowledgement on our contributions. Please feel free to let us know if the reviewer has further questions.

---

### Decision · Program_Chairs · 2025-09-17

**Decision:**

Accept (poster)

**Comment:**

Reviewers agree that this is a significant step forward in the field. The rebuttal and post-rebuttal discussion clarified initial questions. Congratulations, this submission is accepted.